# Neural Low-Discrepancy Sequences

## Abstract

Low-discrepancy points are designed to efficiently fill the space in a uniform manner. This uniformity is highly advantageous in many problems in science and engineering, including in numerical integration, computer vision, machine perception, computer graphics, machine learning, and simulation. Whereas most previous low-discrepancy constructions rely on abstract algebra and number theory, Message-Passing Monte Carlo (MPMC) was recently introduced to exploit machine learning methods for generating point sets with lower discrepancy than previously possible. However, MPMC is limited to generating point sets and cannot be extended to low-discrepancy sequences (LDS), i.e., sequences of points in which every prefix has low discrepancy, a property essential for many applications. To address this limitation, we introduce Neural Low-Discrepancy Sequences (NEUROLDS), the first machine learning-based framework for generating LDS. Drawing inspiration from classical LDS, we train a neural network to map indices to points such that the resulting sequences exhibit minimal discrepancy across all prefixes. To this end, we deploy a two-stage learning process: supervised approximation of classical constructions followed by unsupervised fine-tuning to minimize prefix discrepancies. We demonstrate that NEUROLDS outperforms all previous LDS constructions by a significant margin with respect to discrepancy measures. Moreover, we demonstrate the effectiveness of NEUROLDS across diverse applications, including numerical integration, robot motion planning, and scientific machine learning. These results highlight the promise and broad significance of Neural Low-Discrepancy Sequences.

## 1 Introduction

Approximating integrals using a finite set of sample points is a central task in scientific computation, with applications ranging from numerical integration to uncertainty quantification and Bayesian inference to computer vision and machine learning tasks; (Chen et al., 2018; Paulin et al., 2022; Keller, 2013; Herrmann & Schwab, 2020; Mishra & Rusch, 2021; Longo et al., 2021). Problems that arise in these areas often involve computing expectations of the form $\mathbb{E}_\rho(f)$ of a function $f(\boldsymbol{x})$ in $\mathbb{R}^d$ with respect to some probability distribution $F$ with density function $\rho(\boldsymbol{x})$.

The simple Monte Carlo (MC) method estimates expectations of this kind by drawing samples $\{\mathbf{X}_i\}_{i=1}^N$ randomly IID from $F$ and computing the sample mean, i.e.,

$$\mathbb{E}_\rho(f) = \int_{\mathbb{R}^d} f(\boldsymbol{x})\rho(\boldsymbol{x})\,\mathrm{d}\boldsymbol{x} \approx \frac{1}{N}\sum_{i=1}^N f(\mathbf{X}_i). \tag{1}$$

In this work, we will assume that there exists a transformation to map our integration problem to the $d$-dimensional unit hypercube and thus will assume hereafter that $F$ is the uniform distribution on $[0,1]^d$. Under the assumption that some notion of variance of the integrand $f(\boldsymbol{x})$ is finite (e.g., in the sense of Hardy-Krause; (Owen, 2005)), the standard MC convergence rate of $\mathcal{O}(N^{-1/2})$ applies, which may necessitate very large $N$ when a high degree of accuracy is required. A popular variation on the MC method is to replace the random samples with a deterministic node set that more evenly covers the domain. Such *low-discrepancy* (LD) points form the foundation of quasi-Monte Carlo (QMC) methods; Dick et al. (2013); Hickernell et al. (2025), which achieve error rates close to $\mathcal{O}(N^{-1})$ in favorable cases.

Assuming the integrand belongs to a reproducing kernel Hilbert space (RKHS) $\mathcal{H}$ of functions $\mathbb{R}^d \to \mathbb{R}$ equipped with an inner product $\langle \cdot, \cdot \rangle_{\mathcal{H}}$ and corresponding norm $\| \cdot \|_{\mathcal{H}}$, one can use the Cauchy-Schwarz inequality within $\mathcal{H}$ to derive an error bound on the approximation (1) as

$$\left| \frac{1}{N} \sum_{i=1}^{N} f(\mathbf{X}_i) - \int_{[0,1]^d} f(\boldsymbol{x}) \, \mathrm{d}\boldsymbol{x} \right| \leq \|f\|_{\mathcal{H}} \, D_2^k(\{\mathbf{X}_i\}_{i=1}^N).$$

In the above, $D_2^k(\{\mathbf{X}_i\}_{i=1}^N)$ is referred to as the *kernel discrepancy*, or simply, the *discrepancy* for brevity, and the term $\|f\|_{\mathcal{H}}$ is a measure of variation of the integrand; see Hickernell (1998) for further details. The discrepancy term measures how closely the empirical distribution of the discrete sample point set approximates the uniform distribution on $[0,1]^d$. Denote by $k : \mathbb{R}^d \times \mathbb{R}^d \to \mathbb{R}$ the reproducing kernel associated with the RKHS $\mathcal{H}$. The discrepancy can be computed explicitly as

$$D_2^k(\{\mathbf{X}_i\}_{i=1}^N) = \sqrt{\iint_{[0,1]^d} k(\boldsymbol{x}, \boldsymbol{y}) \, \mathrm{d}\boldsymbol{x} \, \mathrm{d}\boldsymbol{y} - \frac{2}{N} \sum_{i=1}^{N} \int_{[0,1]^d} k(\mathbf{X}_i, \boldsymbol{y}) \, \mathrm{d}\boldsymbol{y} + \frac{1}{N^2} \sum_{i,j=1}^{N} k(\mathbf{X}_i, \mathbf{X}_j)}.$$

$$(2)$$

Thus, employing sampling locations $\{\mathbf{X}_i\}_{i=1}^N$ with small discrepancy, in principle, leads to a more accurate and tighter approximation of the expectation of interest. It then follows that constructing sampling nodes with minimal discrepancy is of broad importance, with applications across many areas of computational science.

## 1.1 SETS VERSUS SEQUENCES

In the study of QMC methods, it is important to distinguish between LD *sets* and *sequences*. Theoretical results on sequences in dimension $d$ often correspond to those on sets in dimension $d + 1$; (Roth, 1954; Kirk, 2020). Thus, despite being closely linked, sets and sequences are designed to address different problems.

LD sets are finite collections of nodes that achieve good uniformity over the $d-$dimensional unit hypercube. For such sets, one can typically establish a discrepancy bound of $C(\log N)^{d-1}/N$ for some absolute constant $C$ depending on the specific construction. These are particularly well suited to applications where the number of samples is known in advance, and this fixed-$N$ setting is exactly the problem addressed by the recently successful Message-Passing Monte Carlo (MPMC) framework; (Rusch et al., 2024). Classical examples of LD sets include Hammersley point sets; (Hammersley, 1960), rank-1 lattices; (Dick et al., 2022), and digital nets; (Dick & Pillichshammer, 2010). In contrast, LD sequences are infinite constructions that are extensible in the number of points, with the property that every initial segment, referred to in this text as a *prefix*, of length $N$ achieves discrepancy of order $\mathcal{O}((\log N)^d/N)$. Classical examples are the van der Corput sequence in one-dimension, Sobol' and Halton sequences for any dimension; (Halton, 1960; Sobol', 1967) and their numerous variants, as well as more modern greedy-optimized extensible constructions; (Kritzinger, 2022). As an important remark, the standard LD sequence constructions typically yield much smaller discrepancy values at special values of $N$, such as a powers of 2, which introduces an inherent limitation in practice as performance can fluctuate depending on the chosen sample size. For example, in the first $2^{14}$ van der Corput points, there is never a better discrepancy for $N$ points than for $2^m$ points when $2^m < N < 2^{m+1}$; (Owen, 2023, Figure 15.4).

An inherent trade-off arises: the discrepancy can often be minimized more effectively for a LD set since the sampling nodes are optimized globally for a particular, pre-specified $N$; (Rusch et al., 2024). However, extending such optimized sets to larger sizes is challenging, as adding new nodes typically disrupts the carefully balanced uniformity. Sequences, on the other hand, provide greater flexibility in practice, since they allow sample sizes to be increased adaptively without restarting the construction, albeit often at the cost of a slightly higher discrepancy for any given $N$. This trade-off is illustrated in Fig. 1, which compares the discrepancy in $d = 4$ of MPMC (trained to minimize the discrepancy at $N = 1024$) with that of classical low-discrepancy sequences as the number of points increases. We observe that MPMC achieves the lowest discrepancy at $N = 1024$, clearly outperforming all other methods at this target length. However, for most $N' < N$, it does not surpass standard low-discrepancy sequences. This is expected, since MPMC was optimized only for the full point set of 1024 points. This highlights the need for a sequential generation mechanism for low-discrepancy sampling points.

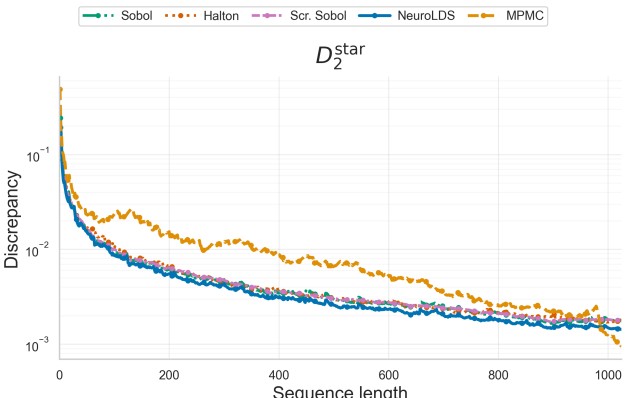

Figure 1: $D_2^{\text{star}}$ discrepancy for increasing number of sampling points in $d = 4$ for MPMC, Sobol', Halton, NEUROLDS, and Scrambled Sobol' (mean over 32 scrambled sequences).

## 1.2 RELATED WORK

There exists a growing literature on the optimization of sampling node locations. Some works aim to minimize the discrepancy directly; (Clément et al., 2025; Mak & Joseph, 2018; Chen et al., 2018; Dwivedi & Mackey, 2024), while others use alterative methods and figures-of-merit; (Miller et al., 2025; Mak & Joseph, 2025; Ding et al., 2025).

The Message-Passing Monte Carlo (MPMC) method; Rusch et al. (2024) is one such general QMC construction method, and unlike classical construction methods such as digital nets or rank-1 lattices, which are derived through number-theoretic means, MPMC is the first work to formulate the problem as one of learning a uniform design through state-of-the-art geometric deep learning tools. The core of the method is a graph neural network based on a message-passing algorithm where each node of the graph corresponds to a sample point and the edges (formed via nearest neighbor) encode the geometric relations between the points. The optimization is initialized by IID random points contained in $[0, 1]^d$ and proceeds by updating the message-passing neural network parameters according to an $L_2$-based discrepancy function–the classical $L_2$-star discrepancy is used in the original formulation–which serves as a differentiable objective function. A resulting transformation is learned which maps the original IID random points into a uniform output set which has very small discrepancy. MPMC has its main strengths in the very effective minimization of discrepancy values for an $N$-element point set–some of the smallest discrepancy values ever obtained–and its flexibility of design via the exchangeability of differentiable loss function including extensions to minimize Stein discrepancies; (Kirk et al., 2025).

Importantly, the limitations of MPMC are also clear: the optimization process must be repeated for each fixed choice of $N$ since the construction is not extensible to larger point sets. Furthermore, the computational cost grows quickly with the number of sampling points and thus training more than a few thousand points becomes very computationally expensive. These limitations highlight the need for a model which allows for sequential and extensible generation of LD *sequences* of points.

## 1.3 OUR CONTRIBUTION

In this paper, following in the footsteps of several recent works employing machine learning architectures in LD optimization pipelines, we develop NEUROLDS, a flexible LDS generator. We further demonstrate that NEUROLDS outperforms classical LD sequences in terms of discrepancy minimization, QMC integration, robot motion planning, and scientific machine learning applications.

## 2 METHODS

NEUROLDS is a deterministic sequence generator $f_\theta : \{1, \ldots, N\} \to [0,1]^d$ that preserves the index-driven construction central to QMC. Classical LD sequences (e.g., Halton and Sobol') are constructed via number-theoretic digit transforms of the index; see Section 2.1 for details. Instead, NEUROLDS feeds the index $i$ through a $K$-band sinusoidal positional encoding and an $L$-layer multi-linear perceptron (MLP) with ReLU and final sigmoid activation functions to obtain $\mathbf{X}_i \in [0,1]^d$. We first carry out a pre-training procedure by approximating a traditional LD sequence (i.e., Sobol') using the mean squared error (MSE), then fine-tune by minimizing closed-form $L_2$-based discrepancy losses over all sequence prefixes; see Clément et al. (2025) and Appendix A for exact expressions of the discrepancy loss functions.

### 2.1 INDEX-BASED SEQUENCE CONSTRUCTION

Classical QMC sequences such as Halton and Sobol' rely on the *index* $i \in \mathbb{N}_0$ as the fundamental input. For Halton, the $j$-th coordinate is obtained from the *radical–inverse* function in base $b_j$ with $b_1, \ldots, b_d$ typically chosen as the first $d$ primes; Halton (1960); Niederreiter (1992); Dick & Pillichshammer (2010). Writing $i = \sum_{k=0}^\infty a_k b_j^k$ with digits $a_k \in \{0, \ldots, b_j - 1\}$, the $j$-th coordinate of the $i$-th point of the Halton sequence is

$$\phi_{b_j}(i) = \sum_{k=0}^\infty a_k\, b_j^{-(k+1)},$$

i.e., take the digits of $i$ in base $b_j$ and place them after the decimal point. The Halton sequence is therefore $\{(\phi_{b_1}(i), \ldots, \phi_{b_d}(i)) : i \geq 0\}$. This yields good equidistribution in low $d$, but number-theoretic correlations emerge as $d$ grows (Niederreiter, 1992; Dick & Pillichshammer, 2010; Kirk & Lemieux, 2025).

Sobol' sequences, in contrast, are digital $(t, d)$-sequences in base 2 constructed from primitive polynomials over the *finite field* $\mathbb{F}_2 := \{0, 1\}$ (arithmetic mod 2; addition is bitwise XOR); Sobol' (1967); Dick & Pillichshammer (2010). Each dimension $j$ uses a polynomial $p_j(z) = z^{m_j} + a_1 z^{m_j - 1} + \cdots + a_{m_j}$ with coefficients $a_i \in \{0, 1\}$ to generate binary *direction numbers* $v_{j,k} \in (0, 1)$ (interpreted as binary fractions). For $k > m_j$, they satisfy

$$v_{j,k} = a_1 v_{j,k-1} \oplus \cdots \oplus a_{m_j} v_{j,k-m_j} \oplus \left(v_{j,k-m_j} \gg m_j\right),$$

where $\oplus$ denotes bitwise XOR and $\gg m_j$ denotes a bitwise right shift by $m_j$ places. Let $g(i) = i \oplus (i \gg 1)$ be the Gray code of $i$, and let $g_k(i)$ be its $k$-th bit. Then

$$X_{i,j} = \bigoplus_{k=1}^\infty g_k(i)\, v_{j,k},$$

interpreted as a binary fraction in $[0, 1)$ by reading the resulting bitstring after the binary point. This digital construction yields very small discrepancy even in moderate $d$ (Dick & Pillichshammer, 2010).

Inspired by these number-theoretic constructions, we also root our approach in the index, but instead of fixed digit expansions or direction numbers, we expose multiple frequency scales of $i$ via sinusoidal features. These features play an analogous role to the radical–inverse digits of Halton or the Gray-coded direction numbers of Sobol', while allowing the downstream MLP to *learn* flexible digital rules according to a discrepancy objective function generating points in $[0, 1]^d$ that achieve very high uniformity. Each index $i$ is mapped to a sinusoidal encoding

$$\psi_i := \psi(i) = \left[\frac{i}{N},\ \sin\!\left(2^k \pi \frac{i}{N}\right),\ \cos\!\left(2^k \pi \frac{i}{N}\right) \ : \ k = 0, \ldots, K-1\right] \in \mathbb{R}^{1+2K},$$

analogous to Fourier features in positional encoding. This embedding exposes multiple frequency scales of the index to the network, mirroring the role of base-$b$ digit expansions in Halton or Sobol'.

The encoded index is passed through an $L$-layer feedforward network with ReLU activations and a final sigmoid, yielding $\mathbf{X}_i = f_\theta(\psi_i) \in [0, 1]^d$. The collection $\{\mathbf{X}_i\}_{i=1}^N$ defines a deterministic, learned sequence.

## 2.2 TWO-STAGE OPTIMIZATION

Training proceeds in two complementary phases, and an overview of the overall architecture is shown in Figure 2.

**Pre-training (MSE alignment).** We initialize $f_\theta$ by regressing onto a chosen reference QMC sequence, with natural starting points being the Sobol' or Halton sequences. Given these targets $\{q_i\}_{i=1}^N$, we minimize

$$\mathcal{L}_{\text{pre}}(\theta) \;=\; \frac{1}{N} \sum_{i=1}^N \big\| f_\theta(\psi_i) - q_i \big\|_2^2.$$

In practice, the reference sequence is generated with an appropriate burn-in period, as is sometimes recommended in the literature; Owen (2022). This pre-training phase stabilizes the learning process and proves essential for the success of our method (see Section 3.3).

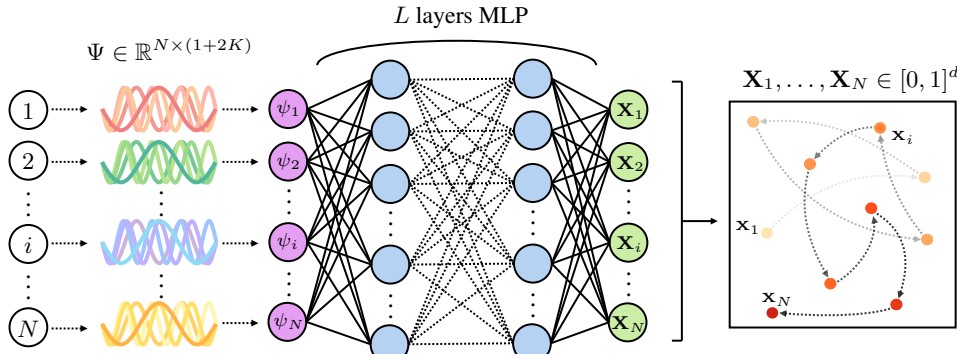

Figure 2: Schematic of the proposed NEUROLDS model. First, each index $i \in \{1, \dots, N\}$ is mapped to a sinusoidal feature vector $\psi_i \in \mathbb{R}^{1+2K}$. Second, the encoded features are passed through an $L$-layer multilayer perceptron (MLP), which outputs $\mathbf{X}_i \in [0,1]^d$. Finally, the collection $\{\mathbf{X}_1, \dots, \mathbf{X}_N\}$ forms a (learned) low-discrepancy sequence within the unit hypercube.

**Fine-tuning (discrepancy minimization).** After pre-training, we further refine $f_\theta$ by minimizing differentiable $L_2$-based discrepancy losses (2), evaluated over all sequence prefixes. These losses are induced by symmetric positive-definite kernels $k$, and we adopt product-form kernels that yield classical discrepancy measures. For example, choosing $k(\boldsymbol{x}, \boldsymbol{y}) = \prod_{j=1}^d \big(1 - \max(x_j, y_j)\big)$ for $\boldsymbol{x}, \boldsymbol{y} \in [0,1]^d$ recovers the standard $L_2$ star discrepancy; Warnock (1972). Other well-studied choices include symmetric, centered, and periodic variants, all of which admit exact and differentiable forms. Each discrepancy can be computed in $\mathcal{O}(dN^2)$ time across all prefixes of a sequence of length $N$. We refer to Clément et al. (2025) for a comprehensive overview of $L_2$ discrepancies, and to Appendix A for explicit kernel definitions, which serve as tunable hyperparameters in NEUROLDS.

In higher dimensions, the efficacy of QMC methods often relies on there being some underlying low-dimensional structure, or decaying importance of variables; Dick et al. (2013); Wang & Sloan (2005). After the emergence of a rigorous framework from weighted function spaces Sloan & Wozniakowski (1998), many works now exist in this setting; Dick et al. (2006); Caflisch & Morokoff (1996); Sloan & Woźniakowski (2001); Gnewuch et al. (2014); Chen et al. (2025). Precisely, one assigns a product weight vector $\boldsymbol{\gamma} = (\gamma_1, \dots, \gamma_d) \in \mathbb{R}_+^d$ often depending on some a-priori knowledge of the problem allowing one to quantify the relative importance of variables. This introduces a tailored training ability of NEUROLDS to be applied to anisotropic integrands, as demonstrated in our Borehole case study in Section 3.2.1.

Our overall fine-tuning loss averages prefix discrepancies:

$$\mathcal{L}_{\text{disc}}(\theta) = \sum_{P=2}^N w_P \cdot D_2^\bullet(\{\mathbf{X}_i\}_{i=1}^P)^2, \tag{3}$$

where $w_P$ are a choice of weights and $\bullet \in \{\text{star}, \text{sym}, \text{ctr}, \text{per}, \text{ext}, \text{asd}\}$ denotes the choice of kernel function.

In this work, all results are produced with uniform weights such that all prefix lengths $P \leq N$ contribute equally to the loss. Alternative weighting schemes and their potential effects are discussed in Section 3.3.

## 3 RESULTS

### 3.1 DISCREPANCY MINIMIZATION

We evaluate the discrepancy of NEUROLDS in dimension $d = 4$ for three different choices of $L_2$-discrepancy losses: the symmetric $D_2^{\text{sym}}$, the star $D_2^{\text{star}}$, and the centered $D_2^{\text{ctr}}$. For each loss, our NEUROLDS sequence is obtained by pretraining on Sobol' prefixes and then fine-tuning on the target discrepancy. To ensure comparable conditions, we apply a burn-in of 128 points across the board: (i) during pretraining we regress to Sobol' with the first 128 points discarded (i.e., indices are shifted by 128), and (ii) Sobol' and Halton baselines are likewise evaluated after discarding their first 128 points. Hyperparameters for NEUROLDS are selected *per loss* using Optuna; (Akiba et al., 2019); the best configurations are summarized in the Appendix in Table 5.

Besides the classical Sobol' and Halton baselines, we also compare against randomized Sobol' via Owen's nested uniform scrambling; (Owen, 1995; 1997). This yields sequences that retain low-discrepancy guarantees in expectation, while breaking any uniformity flaws. In our experiments, we report the mean over 32 independent scramblings.

Table 6 in the Appendix reports the exact discrepancy values for several values of $N$. NEUROLDS achieves the lowest discrepancy at every sequence length for all three losses. Scrambling reduces some oscillations in Sobol', but even its randomized variants remain above NEUROLDS for all reported $N$. The full sequence discrepancy profiles are shown in Fig. 3. For completeness, the Appendix also reports the corresponding $D_2^{\text{sym}}$ discrepancy curves in dimensions 1–3 (see Fig. 10), which serve as a lower-dimensional reference and show that improvements are modest in 1D but become progressively clearer as dimension increases.

For visual aid, Fig. 6 in the Appendix shows the first $N \in \{64, 128, 256\}$ points of Sobol' and NEUROLDS in two dimensions. For this illustrative experiment, NEUROLDS was trained with the $D_2^{\text{sym}}$, and all hyperparameters were tuned via Optuna. Although Sobol' achieves good global coverage, it often exhibits structured alignments and mild clustering artifacts, particularly at small $N$. By contrast, NEUROLDS points appear more irregular, or random, while still covering the domain evenly.

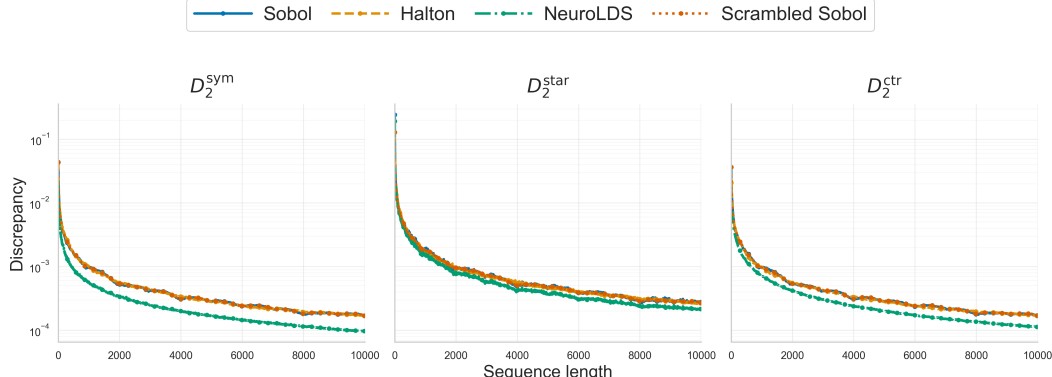

Figure 3: Comparison plots of different discrepancy metrics for 4d point sequences up to a length of 10000 as a function of sequence length: $D_2^{\text{sym}}$ (left), $D_2^{\text{star}}$ (middle), and $D_2^{\text{ctr}}$ (right). Each panel compares Sobol', Halton, NEUROLDS, and Scrambled Sobol' (mean over 32 scrambled sequences).

Table 1: Absolute errors for the Borehole integral approximation with increasing sequence length ($N$) for NEUROLDS, Sobol', Halton and a greedy construction. Bold marks the best error per row, underlined the second best.

| $N$ | NEUROLDS (ours) | Sobol' | Halton | NM-Greedy |
|-----|-----------------|--------|--------|-----------|
| 20  | **1.0309** | 7.3049 | 3.3671 | 3.8426 |
| 60  | 0.1864 | **0.1840** | 0.7724 | 0.5877 |
| 100 | 0.5765 | 0.4879 | 1.1391 | **0.0299** |
| 140 | **0.2095** | 0.4487 | 0.8304 | 0.2601 |
| 180 | **0.4204** | 0.4222 | 1.0487 | 0.9011 |
| 220 | 0.3430 | **0.2611** | 1.1711 | 0.5779 |
| 260 | **0.0239** | 0.5665 | 0.4516 | 0.7185 |
| 300 | **0.0467** | 0.0951 | 0.5534 | 0.0726 |
| 340 | **0.1478** | 0.2806 | 0.6215 | 0.6954 |
| 380 | 0.2059 | **0.1129** | 0.1613 | 0.4612 |
| 420 | **0.1066** | 0.1837 | 0.2260 | 0.4224 |
| 460 | 0.0657 | 0.1086 | 0.2164 | **0.0587** |
| 500 | **0.0651** | 0.1121 | 0.2619 | 0.2205 |

## 3.2 APPLICATIONS

### 3.2.1 QUASI-MONTE CARLO INTEGRATION

LDS are most often employed in QMC integration to approximate a $d$-dimensional integral over $[0, 1]^d$, with the goal of achieving faster convergence than standard Monte Carlo. To illustrate the effectiveness of NEUROLDS in this setting, we consider the 8-dimensional Borehole function, a benchmark in uncertainty quantification, sensitivity analysis, and approximation algorithms (An & Owen, 2001; Morris et al., 1993; Kersaudy et al., 2015). The function models the flow rate of water through a borehole connecting two aquifers separated by an impermeable rock layer. Our objective is to approximate the expected flow rate, i.e., the integral of the Borehole function over the 8-dimensional parameter space; see Appendix B for details.

Since no analytical solution is available, we pre-compute a high-fidelity reference value using plain Monte Carlo with $N = 2^{21}$ samples. Against this benchmark, we compare several low-discrepancy constructions: classical Sobol' and Halton sequences, our proposed NEUROLDS, and a greedy baseline obtained via Nelder–Mead optimization (Nelder & Mead, 1965); see (Chen et al., 2018, Appendix B.2.1) for implementation details. Both NEUROLDS and the NM-Greedy construction are optimized against the weighted symmetric $L_2$ discrepancy (see Appendix A), using a computed weight vector $\boldsymbol{\gamma}$ that down-weights irrelevant coordinates identified by sensitivity analysis and penalizes irregularity more evenly across dimensions; details are again given in Appendix B. A visualization of the weighted star–discrepancy behaviour for all methods is provided in Appendix 12.

The experimental protocol is deterministic: for each sequence type we generate a single fixed instance of the first $N$ points, without randomization or scrambling, and compute the absolute error of the sample mean estimator of the Borehole integral. Table 1 reports these errors across a range of $N$, with the smallest error in each row highlighted in bold.

We find that NEUROLDS, trained with coordinate-weighted discrepancy losses to emphasize the most influential variables, consistently delivers superior accuracy compared with the other sequences across nearly all values of $N$. By contrast, Halton performs the worst overall, as expected given its well-documented uniformity pathologies in moderate to high dimensions (Kirk & Lemieux, 2025). Although the NM-Greedy construction also incorporates sensitivity-informed weights, its performance lags behind NEUROLDS. Notably, NEUROLDS achieves either the smallest or second-smallest error for all but two tested values of $N$.

### 3.2.2 ROBOT MOTION PLANNING

Optimal exploration of the configuration space is a crucial component of sampling based motion planning. Indeed, if the sampled points leave large gaps or cluster unevenly, the planner may fail to discover important paths between regions, particularly when narrow passages are present. For

Table 2: RRT success rates (in %) for NEUROLDS, Sobol, Halton, and Uniform sampling. Each distribution uses 10 precomputed sequences of length $N = 10^5$, reused across random initializations and difficulty levels. Bold marks the best per row, underline the second best.

| Experiment | Passage width | NEUROLDS (ours) | Sobol' | Halton | Uniform |
|---|---|---|---|---|---|
| 4d Kinematic Chain | 0.64 | **96.58** | 73.04 | 87.95 | 85.83 |
| | 0.60 | **94.41** | 72.29 | 86.95 | 76.58 |
| | 0.56 | 83.60 | 73.41 | **83.66** | 68.88 |
| | 0.52 | 79.87 | 68.69 | **82.54** | 71.61 |
| | 0.48 | **80.06** | 68.75 | 72.85 | 69.06 |
| | 0.44 | **73.16** | 64.96 | 71.24 | 72.23 |
| | 0.40 | **80.00** | 65.59 | 67.32 | 68.38 |

a specific family of planners-Probabilistic Roadmaps (PRM)-there exist theoretical guarantees that directly connect the quality of sampling to path optimality (Janson et al., 2018). More recently, the guarantee has been expressed in terms of discrepancy, and significant empirical performance gains obtained with MPMC *point sets* on a suite of PRM benchmarks (Chahine et al., 2025).

However, most sampling-based planners construct their search structures *sequentially*. For example, the Rapidly-exploring Random Tree (RRT) grows strictly one node at a time, with each new sample extending the current frontier of exploration (Algorithm provided in Appendix C). In this setting, order matters significantly: an early bias toward one region may starve others, leading to poor coverage of narrow passages. The sequential sampling structure must align with the incremental expansion of RRT, with both spatial distribution and temporal ordering central to performance.

We compare the sampling quality of NEUROLDS against uniform sampling, as well as Halton and Sobol' sequences, on a challenging motion planning experiment: *Kinematic Chain in a Semi-Circular Tunnel*. The task is to control a chain of 4 revolute joints in the plane from an initial configuration inside a semi-circular tunnel to an exterior target pose. The challenge lies in threading the articulated links through the curved tunnel, requiring coordinated rotations across all joints. We randomize the placement of the tunnel between runs, with 160 repetitions per sequence. The success rates for the experiment under different difficulty levels are summarized in Table 2. Across all passage widths, NEUROLDS consistently delivers the highest success, demonstrating its ability to guide the planner through narrow passages. Moreover, when we fix the average success rate (aggregated over all passage widths) achieved by NeuroLDS, we find that Sobol' requires 2.50 times as many sampling points, Halton requires 1.55 times as many, and Uniform sampling requires 2.27 times as many to reach the same average performance. The structured, adaptive sampling of NeuroLDS produces well-distributed points that efficiently cover the configuration space, ensuring robust path discoveries. Overall, these results highlight the advantage of neural-adapted sampling in motion planning tasks, where sequential exploration and effective coverage are critical.

### 3.2.3 SCIENTIFIC MACHINE LEARNING

LDS have been shown to improve the generalization of deep neural networks trained to approximate parameter-to-observable mappings arising from parametric partial differential equations (PDEs) (Mishra & Rusch, 2021; Longo et al., 2021). In this section, we empirically test whether NEUROLDS improves the performance of deep neural networks over random points and competing LDS constructions in this context. To this end, we consider a multidimensional Black-Scholes PDE that models the pricing of a European multi-asset basket call option, where the assets in the basket are assumed to change in time according to a multivariate geometric brownian motion. The details of the PDE is given in Appendix D. In our experimental setup, we train neural networks to predict the 'fair price', i.e., solution of the Black-Scholes PDE at time $t = 0$, for different initial values of the underlying asset prices drawn from $[0, 1]^2$. We compare the performance of NEUROLDS against uniform random points as well as Sobol' sequences and NMGreedy. We fix the length of the sequence to 1000. To ensure meaningful results, we present the average of 20 training runs of the same network using different random weight initializations. Moreover, we conduct a light random search to optimize the learning rate, width, and number of layers of the trained MLP for each of the

three sampling methods. The results can be seen in Table 3. While Sobol' sequences outperform uniform random points and NM Greedy falls short, our NEUROLDS achieve superior performance over the baselines. Moreover, NEUROLDS achieves lower error than the baselines for any choice of the $L_2$-discrepancy. We conclude that NEUROLDS can successfully be leveraged in the context of scientific machine learning.

Table 3: Average mean-squared error (MSE) of fair price predictions for the 2d Black–Scholes PDE over 20 runs (lower is better). Values are reported as $[\times 10^{-4}]$. Best result in bold.

| | Baselines | | | NEUROLDS (ours) | | | | | |
|---|---|---|---|---|---|---|---|---|---|
| | MC | Sobol' | NMGreedy | $D_2^{\text{sym}}$ | $D_2^{\text{star}}$ | $D_2^{\text{ext}}$ | $D_2^{\text{per}}$ | $D_2^{\text{ctr}}$ | $D_2^{\text{asd}}$ |
| MSE $[\times 10^{-4}]$ | 4.24 | 4.04 | 4.72 | 3.69 | 3.66 | 4.01 | 3.82 | **3.34** | 3.63 |

## 3.3 MODEL ABLATIONS AND SENSITIVITY STUDIES

We now examine the role of individual design choices in NEUROLDS.

**Pre-training vs. direct discrepancy minimization.** To assess whether pre-training is necessary, we compared two pipelines *in 2d, 3d, and 4d*: 1) *Pre-train+FT:* regress to Sobol' sequence and then fine-tune on the target discrepancy; 2) *Direct:* initialize randomly and optimize the discrepancy loss from scratch. We kept the architecture, optimizer, and budget fixed, and tuned hyperparameters for each pipeline via Optuna. Across all dimensions, the *Direct* variant consistently collapsed to a degenerate solution in which points concentrate near a corner of $[0,1]^d$ (a pathology observed also in related work Clément et al. (2025)), while *Pre-train+FT* remained stable. This indicates that Sobol'-based pre-training provides an essential inductive bias before discrepancy refinement.

**Alternative sequence generation with autoregressive GNNs.** To test if explicit autoregression helps, we implemented an AR-GNN that emits one point conditioned on the previous prefix and compared it to our index-conditioned MLP *in 2d*. We matched parameter count and training budget and tuned both models with Optuna. While the AR-GNN performed reasonably for small sequence lengths, beyond a few hundred points the training signal became too weak to propagate through long contexts without truncation, and performance degraded relative to the index-based MLP. Because truncation undermines global low-discrepancy structure, we adopt the simpler index-based formulation, which proved more robust across dimensions.

**Effect of sinusoidal frequency parameter $K$.** To isolate the contribution of the index encoding bandwidth, we trained models *in 2d, 3d, and 4d* that are *identical in every respect except $K$*, using $K \in \{8, 16, 32\}$ (Optuna re-tunes learning-rate and weight-decay per $K$). In 4d, the ablation is visualized in the dedicated panel figure (see Fig. 7): larger $K$ reduces fluctuations of $D_2^{\text{sym}}$ (i.e., more stable curves), whereas small $K$ (e.g., 8) exhibits higher variance. Thus, increasing $K$ improves stability of the discrepancy profile at the cost of a modest increase in training time.

**Impact of non-linearities.** We evaluated the necessity of depth by replacing the MLP with a purely linear map (followed by a sigmoid) and by a shallow one-hidden-layer ReLU MLP, all *in 2d and 3d* with Optuna-tuned hyperparameters. The linear model systematically failed to approximate Sobol'-like structure and yielded poor discrepancy throughout the prefix. A single hidden layer could reach competitive levels but required substantially longer training time. Our deeper MLP strikes a favorable accuracy–efficiency balance across dimensions.

**Impact of weights in fine-tuning discrepancy loss.** We run on $d = 4$ with 10000 points and optimal hyperparameters tuned via Optuna, and compare the effect of using uniform weighting $w_P = 1/(N-2)$ versus length-proportional weighting $w_{P*} = 2P/(N^2 + N - 2)$. Clearly, from Figure 8 from the Appendix we see that performance is comparable: the uniform version performs slightly better for shorter prefixes, while $w_{P*}$ yields lower discrepancy on longer prefixes. This matches intuition, since $w_{P*}$ places more emphasis on later prefixes, thus optimizing them at the expense of early ones.

**Replacing the MLP with an LSTM.** To assess whether recurrence offers any benefit for low-discrepancy sequence generation, we replace the index-conditioned MLP with a single-layer LSTM and evaluated both architectures for $d = 4$ under the $D_2^{\text{star}}$ objective with a target length of $500$ points. Hyperparameters for each model were tuned independently with Optuna, and Fig. 9 reports the best-performing trial from each sweep. In this controlled comparison, the LSTM's best configuration achieves a slightly lower discrepancy curve than the best MLP run, indicating that recurrent state can, at least in principle, capture marginally richer structure from the index embedding. However, this comes at a substantial computational cost: over five repeated sweeps, full MLP optimization requires roughly ten minutes on average, whereas the LSTM exceeds one hour. Because this gap grows even further for longer sequences, training an LSTM for 10,000 points becomes prohibitively expensive, especially given that the modest improvement it provides does not translate into any practical advantage. Consequently, the simpler MLP remains the more efficient architecture for NEUROLDS.

## 4 DISCUSSION

NEUROLDS shows that neural architectures can successfully generate LDS, providing a solution to the fixed $N$ limitation of the MPMC model (Rusch et al., 2024), with strong empirical performance across numerical integration, path planning and scientific machine learning tasks. Our model formulation as presented here focuses on classical $L_2$-based discrepancy functions to target the uniform distribution on $[0,1]^d$, the traditional setting for QMC methods. However, we note that the presented framework is flexible, and can be extended to more general notions of kernel discrepancies, such as Stein discrepancies (Liu et al., 2016), enabling designs that compact non-uniform distributions into an extensible sequence of nodes.

Finally, we note that our reliance on Sobol' or Halton pre-alignment underlines the continued importance of classical number-theoretic and non-ML optimization-based QMC constructions, which provide the stability needed for successful training as highlighted in Section 3.3.

## 5 REPRODUCIBILITY STATEMENT

We provide an anonymized supplementary code package that implements our method to *generate low-discrepancy sequences via NeuroLDS*. A public repository with identical code will be released after review.

## 6 THE USE OF LARGE LANGUAGE MODELS (LLMS)

LLMs were used for language rephrasing and light coding assistance (e.g., refactoring, boilerplate, docstrings, linting fixes). They did not originate methods, model designs, or results; the authors take full responsibility. LLMs are not eligible for authorship.

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

**Supplementary Material for:**
Neural Low-Discrepancy Sequences

## A  THE DISCREPANCY AND ASSOCIATED KERNELS

In the main text, it is discussed how several variants of the discrepancy can be formulated by substituting an appropriate symmetric and positive-definite kernel into equation (2). In this Appendix, we list popular choices of (product) kernels and the associated discrepancy function. The interested reader is additionally referred to Clément et al. (2025).

Table 4: Kernel functions corresponding to commonly used $L_2$ discrepancy measures.

| Discrepancy | Kernel function $k(\boldsymbol{x}, \boldsymbol{y})$ |
|---|---|
| Star ($D_2^{\text{star}}$) | $\prod_{j=1}^{d} \big(1 - \max(x_j, y_j)\big)$ |
| Extreme ($D_2^{\text{ext}}$) | $\prod_{j=1}^{d} \Big(\min(x_j, y_j) - x_j y_j\Big)$ |
| Periodic ($D_2^{\text{per}}$) | $\prod_{j=1}^{d} \Big(\frac{1}{2} - |x_j - y_j| + (x_j - y_j)^2\Big)$ |
| Centered ($D_2^{\text{ctr}}$) | $\prod_{j=1}^{d} \frac{1}{2}\Big(|x_j - \frac{1}{2}| + |y_j - \frac{1}{2}| - |x_j - y_j|\Big)$ |
| Symmetric ($D_2^{\text{sym}}$) | $\prod_{j=1}^{d} \frac{1}{4}\big(1 - 2|x_j - y_j|\big)$ |
| Average squared ($D_2^{\text{asd}}$) | $\prod_{j=1}^{d} \frac{1}{2}\big(1 - |x_j - y_j|\big)$ |

In higher dimensions, it is recommended to introduce coordinate weights $\boldsymbol{\gamma} \in \mathbb{R}_+^d$ to reflect variable importance. In this weighted setting, kernels take the form

$$\widetilde{k}(\boldsymbol{x}, \boldsymbol{y}) \; = \; \prod_{j=1}^{d} \Big(1 + \gamma_j \, k(x_j, y_j)\Big),$$

where $k(\cdot, \cdot)$ denotes a one-dimensional kernel selected from Table 4.

## B  ON THE BOREHOLE FUNCTION

The Borehole function models the steady-state water flow rate through a borehole that connects two aquifers separated by an impermeable rock layer. Mathematically, the function is:

$$f(r_w, r, T_u, H_u, T_l, H_l, L, K_w) \; = \; \frac{2\pi T_u \, (H_u - H_l)}{\ln\!\left(\frac{r}{r_w}\right) \left(1 + \frac{2L T_u}{\ln(r/r_w) \, r_w^2 K_w} + \frac{T_u}{T_l}\right)}.$$

The input parameters are taken to be independent and uniformly distributed over the following ranges:

$$r_w \in [0.05, \ 0.15] \qquad \text{(borehole radius)}$$
$$r \in [100, \ 50000] \qquad \text{(radius of influence)}$$
$$T_u \in [63070, \ 115600] \qquad \text{(transmissivity of upper aquifer)}$$
$$H_u \in [990, \ 1110] \qquad \text{(potentiometric head of upper aquifer)}$$
$$T_l \in [63.1, \ 116] \qquad \text{(transmissivity of lower aquifer)}$$
$$H_l \in [700, \ 820] \qquad \text{(potentiometric head of lower aquifer)}$$
$$L \in [1120, \ 1680] \qquad \text{(length of borehole)}$$
$$K_w \in [9855, \ 12045] \qquad \text{(hydraulic conductivity of borehole)}$$

**Sensitivity Analysis.** To justify the choice of coordinate weight vector, we performed a global sensitivity analysis of the Borehole function using Sobol' indices.

We employed the Saltelli sampling scheme (Saltelli, 2002; Saltelli et al., 2010) as implemented in the SALib package (Herman & Usher, 2017). For a problem of dimension $d$, this procedure requires $N(2d + 2)$ model evaluations. The outputs were then analyzed to compute: the first-order Sobol' index $S_i$ measuring the proportion of output variance attributable to variations in input $x_i$ alone, and the total-order index $S_{Ti}$ capturing the variance due to $x_i$ including all its interactions with other variables.

Figure 4 reveals that the borehole radius $r_w$ dominates the model response, with $S_1 \approx 0.83$ and $S_{T1} \approx 0.87$. Other parameters, such as $H_u, H_l$ and $L$ contribute at an order of magnitude smaller level, while the remaining variables have negligible influence.

On this basis, we constructed a coordinate weight vector that assigns the highest weight to $r_w$, reduced weights to $H_u, H_l$ and $L$ and near-zero weights to the remaining variables. Specifically, the Sobol' indices were normalized against the maximum index, and then mapping these values (with a small floor added) into product weights resulting in $\gamma = (1.0000, 0.0010, 0.0010, 0.0633, 0.0010, 0.0634, 0.0610, 0.0158)$.

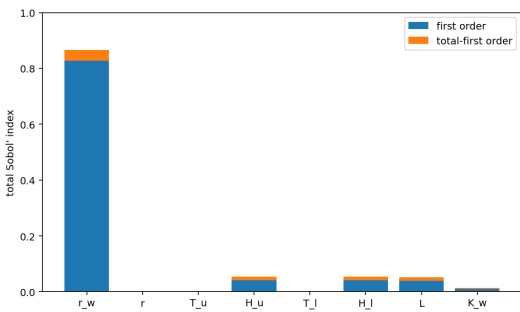

Figure 4: Sobol' indices for the Borehole function.

## C  ON RAPIDLY-EXPLORING RANDOM TREES

Rapidly-exploring Random Trees (RRT) are a widely used class of sampling-based motion planners designed to efficiently explore high-dimensional configuration spaces. They are particularly effective in tasks with complex constraints or narrow passages, where uniform sampling alone may fail to discover feasible paths. At each iteration, RRT incrementally grows a tree rooted at the initial configuration by sampling a new configuration, extending the nearest tree node toward it, and checking for collisions. The tree grows sequentially, which makes both the spatial distribution and temporal order of the samples critical to planner performance.

Algorithm 1 summarizes the standard RRT procedure. The main parameters include the maximum number of iterations $K$ and the step size $\delta$, which control the exploration extent and granularity of the tree.

---

**Algorithm 1** Rapidly-exploring Random Tree (RRT)

---

**Input:** Initial configuration $x_{\text{init}}$, goal region $\mathcal{X}_{\text{goal}}$, maximum iterations $K$, step size $\delta$

1: Initialize tree $T$ with root $x_{\text{init}}$
2: **for** $k = 1$ to $K$ **do**
3:     Sample a random configuration $x_{\text{rand}}$ from the configuration space $\mathcal{X}$
4:     Find nearest node $x_{\text{nearest}}$ in $T$ to $x_{\text{rand}}$
5:     Compute a new node $x_{\text{new}}$ by moving from $x_{\text{nearest}}$ toward $x_{\text{rand}}$ by step size $\delta$
6:     **if** $x_{\text{new}}$ is collision-free **then**
7:         Add $x_{\text{new}}$ to $T$ with an edge from $x_{\text{nearest}}$
8:         **if** $x_{\text{new}} \in \mathcal{X}_{\text{goal}}$ **then**
9:             **return** Path from $x_{\text{init}}$ to $x_{\text{new}}$
10:         **end if**
11:     **end if**
12: **end for**
13: **return** Failure (no path found)

---

Figure 5 illustrates the Kinematic chain experiment. The panel shows a 4d kinematic chain in a semi-circular tunnel, which we use as our test scenario. Even in the 4d case with 10k sampled points, controlling all four joint angles simultaneously is already non-trivial, highlighting the need for well-structured sampling sequences.

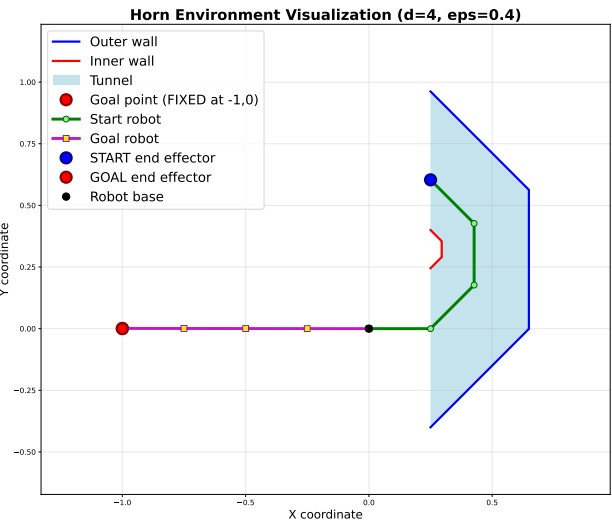

Figure 5: Visualization of kinematic chain task for a 4d chain in a semi-circular tunnel.

## D  ON THE BLACK-SCHOLES PDE EXAMPLE

We consider the following Black-Scholes partial differential equation (BSPDE):

$$\frac{\partial V}{\partial t} + \sum_{i=1}^{d} r S_i \frac{\partial V}{\partial S_i} + \frac{1}{2} \sum_{i=1}^{d} \sigma_i^2 S_i^2 \frac{\partial^2 V}{\partial S_i^2} + \sum_{i=1}^{d-1} \sum_{j=i+1}^{d} \rho_{ij} \sigma_i \sigma_j S_i S_j \frac{\partial^2 V}{\partial S_i \partial S_j} - rV = 0, \quad (4)$$

subject to the terminal condition $V(T, \mathbf{S}) = \Lambda(\mathbf{S})$. Moreover, we consider the payoff function of a European geometric average basket call option:

$$\Lambda(\mathbf{S}(T)) = \max\left\{ \left( \prod_{i=1}^{d} S_i(T) \right)^{1/d} - K,\, 0 \right\}. \quad (5)$$

Since products of log-normal random variables are themselves log-normal, the initial value of the pricing function—i.e., the discounted expectation at time $t = 0$—is

$$V(t, \mathbf{S}) = e^{-r(T-t)}\mathbb{E}_{\tilde{P}}[\Lambda(\mathbf{S}(T)) \mid \mathbf{S}(t) = \mathbf{S}].$$

This quantity is the option's value at purchase time, i.e., the fair price of the European geometric average basket call option under the Black–Scholes model. Using (5), the closed-form solution to the multidimensional BSPDE (4) at $(0, \mathbf{S})$ is (Theorem 2 in Korn & Zeytun (2013)):

$$V(0, \mathbf{S}) = e^{-rT}\big(\tilde{s}\,e^{\tilde{m}}\Phi(d_1) - K\,\Phi(d_2)\big), \tag{6}$$

where $\Phi$ is the standard normal CDF, and

$$\nu = \frac{1}{d}\sqrt{\sum_{j=1}^{d}\left(\sum_{i=1}^{d}\sigma_{ij}^2\right)^2}, \qquad m = rT - \frac{T}{2d}\sum_{i=1}^{d}\sum_{j=1}^{d}\sigma_{ij}^2, \qquad \tilde{m} = m + \frac{1}{2}\nu^2,$$

$$\tilde{s} = \left(\prod_{i=1}^{d}S_i\right)^{1/d}, \qquad d_1 = \frac{\log(\tilde{s}/K) + m + \nu^2}{\nu}, \qquad d_2 = d_1 - \nu,$$

with $\sigma \in \mathbb{R}^{d\times d}$ the covariance matrix of stock returns (entries $\sigma_{ij}$). For the experiments below we fix $\sigma = 10^{-5}\mathbf{I}$, $T = 5$, $K = 0.08$, and $r = 0.05$, where $\mathbf{I} \in \mathbb{R}^{d\times d}$ is the identity matrix.

## E  ADDITIONAL TABLES AND PLOTS

This section provides supplementary material in the form of additional plots and tables that complement the main results. Figure 7 illustrates the behavior of NEUROLDS discrepancy under different settings of $K$, highlighting both the overall trend and the fluctuation indices across prefix lengths. Tables 5 and 6 report quantitative comparisons of prefix discrepancies against standard baselines as well as the best hyperparameters found via Optuna. Together, these results give a more detailed view of model performance and robustness beyond the main text.

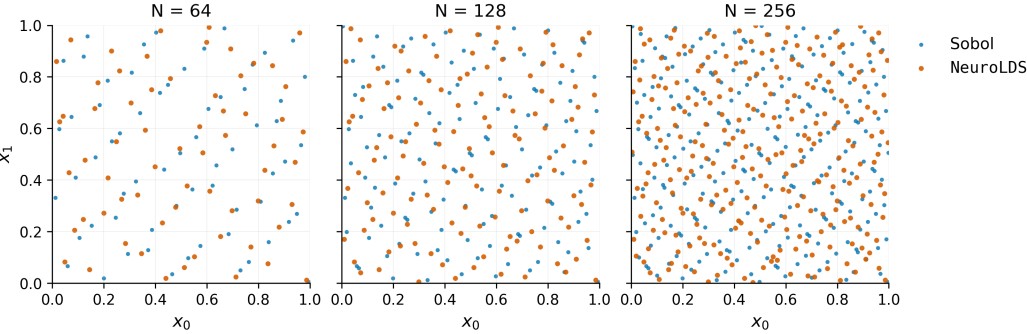

Figure 6: First $N \in \{64, 128, 256\}$ points of Sobol' (blue) and NEUROLDS (orange) in 2d. NEUROLDS was trained with the $D_2^{\text{star}}$ discrepancy and Optuna-tuned hyperparameters. Sobol' exhibits visible structure and clustering, while NEUROLDS distributes points more irregularly yet evenly, mitigating alignment artifacts and leading to lower prefix discrepancy.

Table 5: Optuna-selected hyperparameters per loss (rounded). All runs use sequence length $10^4$, pretrained on Sobol' with 128 burn-in points, then fine-tuned on the target loss.

| Loss | Hidden | Layers | $K$ | Pretrain LR | Fine-tune LR | Final LR ratio |
|------|--------|--------|-----|-------------|--------------|----------------|
| $D_2^{\text{sym}}$ | 768 | 7 | 64 | $2.61\times10^{-3}$ | $5.04\times10^{-3}$ | $3.02\times10^{-2}$ |
| $D_2^{\text{star}}$ | 512 | 5 | 64 | $1.38\times10^{-3}$ | $3.52\times10^{-4}$ | $4.39\times10^{-2}$ |
| $D_2^{\text{ctr}}$ | 768 | 7 | 32 | $2.85\times10^{-3}$ | $4.14\times10^{-3}$ | $1.14\times10^{-1}$ |

Table 6: Prefix discrepancy in $d = 4$ at lengths $N \in \{100, 500, 1000, 2000, 5000, 10000\}$ for our model, Sobol', Halton, and the mean over 32 scrambled Sobol' sequences. Lower is better; best per loss/length in bold.

| | $D_2^{\text{sym}}$ | | | |
| --- | --- | --- | --- | --- |
| **Seq. Length** | **Model** | Sobol' | Halton | Scr. Sobol' (mean) |
| 100 | **0.002669** | 0.004840 | 0.005020 | 0.004602 |
| 500 | **0.000900** | 0.001615 | 0.001608 | 0.001610 |
| 1000 | **0.000578** | 0.000972 | 0.001002 | 0.000965 |
| 2000 | **0.000339** | 0.000527 | 0.000550 | 0.000540 |
| 5000 | **0.000167** | 0.000282 | 0.000278 | 0.000282 |
| 10000 | **0.000098** | 0.000167 | 0.000168 | 0.000169 |
| | $D_2^{\text{star}}$ | | | |
| **Seq. Length** | **Model** | Sobol' | Halton | Scr. Sobol' (mean) |
| 100 | **0.008603** | 0.009477 | 0.010333 | 0.010033 |
| 500 | **0.002585** | 0.002977 | 0.003052 | 0.003058 |
| 1000 | **0.001491** | 0.001824 | 0.001709 | 0.001818 |
| 2000 | **0.000776** | 0.000929 | 0.000937 | 0.000962 |
| 5000 | **0.000368** | 0.000462 | 0.000459 | 0.000460 |
| 10000 | **0.000213** | 0.000266 | 0.000266 | 0.000268 |
| | $D_2^{\text{ctr}}$ | | | |
| **Seq. Length** | **Model** | Sobol' | Halton | Scr. Sobol' (mean) |
| 100 | **0.003534** | 0.004607 | 0.004612 | 0.004673 |
| 500 | **0.001192** | 0.001654 | 0.001531 | 0.001621 |
| 1000 | **0.000711** | 0.000987 | 0.000988 | 0.000958 |
| 2000 | **0.000419** | 0.000541 | 0.000554 | 0.000539 |
| 5000 | **0.000200** | 0.000281 | 0.000279 | 0.000283 |
| 10000 | **0.000114** | 0.000167 | 0.000167 | 0.000169 |

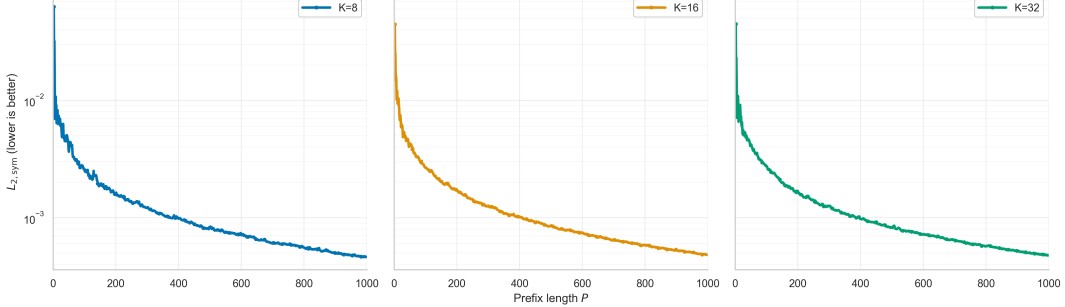

Figure 7: NeuroLDS discrepancy under $D_2^{\text{sym}}$ in 4d as a function of prefix length $P$. The three panels show $K \in \{8, 16, 32\}$ (one per panel). Models differ only in $K$; all remaining hyperparameters and the training setup are identical. The $y$-axis is log-scaled; lower is better. To quantify fluctuations across prefixes, we report a *fluctuation index* $\rho_K := \text{CV}[D_P] = \text{std}(D_P)/\text{mean}(D_P)$ computed over $P \leq 10^4$, where $D_P$ denotes the $D_2^{\text{sym}}$ discrepancy at prefix $P$. We obtain $\rho_8 = 1.886$, $\rho_{16} = 1.608$, and $\rho_{32} = 1.537$ (smaller is smoother). For reference, the 90% log-amplitude $\Delta_K^{90} := \text{P95}[\log_{10} D_P] - \text{P5}[\log_{10} D_P]$ equals 0.954, 0.915, and 0.916 for $K = 8, 16, 32$, respectively.

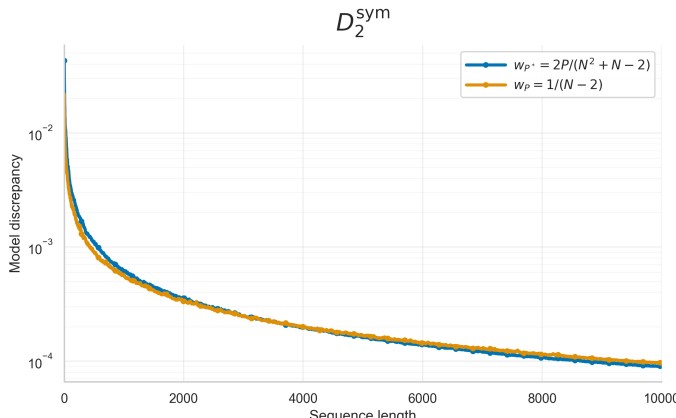

Figure 8: Comparison of fine-tuning with uniform weights ($w_P = 1/(N-2)$) versus length-proportional weights ($w_{P*} = 2P/(N^2 + N - 2)$) in 4d with $N$=10,000 points. Uniform weights favor early prefixes, whereas $w_{P*}$ improves long-prefix performance.

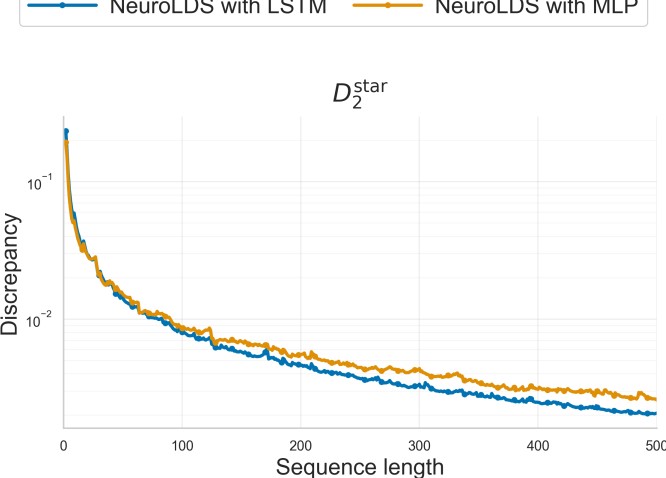

Figure 9: Comparison between LSTM and MLP in 4D under the $D_2^\star$ objective for 500 points. The LSTM achieves a slightly lower discrepancy curve, but reaching this configuration requires over one hour of optimization on average, compared to approximately ten minutes for the MLP. This substantial cost difference grows with sequence length, making the recurrent model impractical for larger prefix budgets.

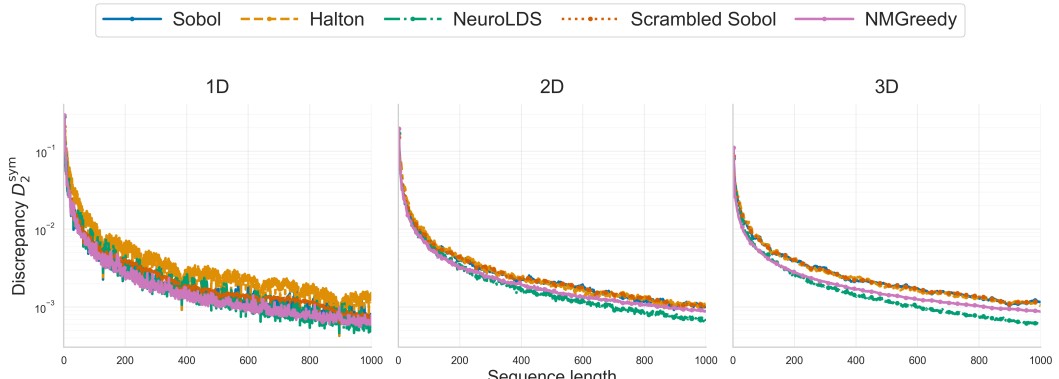

Figure 10: Comparison of $D_2^{\mathrm{sym}}$ discrepancy in 1D, 2D, and 3D. Sequences of length 1000 generated by Sobol', Halton, Scrambled Sobol', NMGreedy and NEUROLDS are shown across three dimensions. In 1D, all constructions achieve very similar discrepancy profiles, with only minor fluctuations separating them. In 2D and 3D, NEUROLDS exhibits a clearer advantage, maintaining lower discrepancy throughout the prefix compared to classical LDS baselines. The figure highlights that benefits of the learned generator become more pronounced as dimensionality increases.

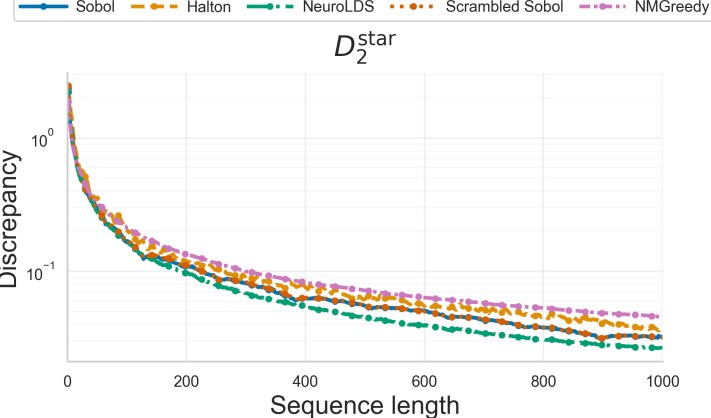

Figure 11: The plot reports the weighted $D_2^{\star}$ discrepancy with $\gamma = (1, 1, 1, 1, 1, 1, 1, 1)$ for Sobol', Halton, Scrambled Sobol', NMGreedy, and NEUROLDS for $d = 8$, up to $N = 1000$ points. NEUROLDS consistently attains the lowest discrepancy across most prefix lengths, while NMGreedy performs worst overall. Scrambled Sobol' reduces some of the oscillations of classical Sobol' but still remains above NEUROLDS throughout the range.

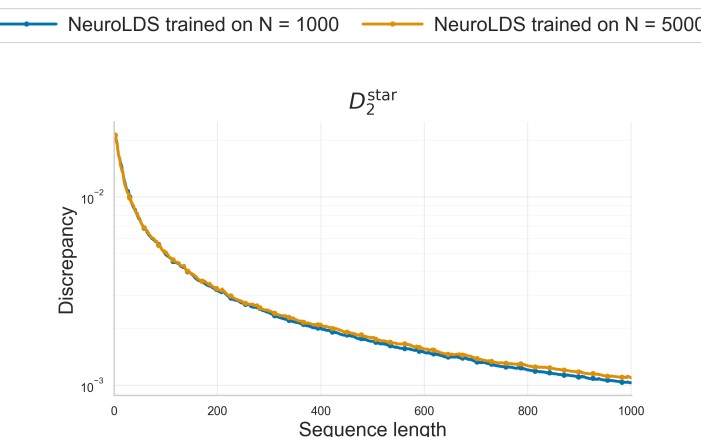

Figure 12: $D_2^\star$ discrepancy for two NeuroLDS for increasing number of points. One NeuroLDS is trained on $N = 1000$ points, while the other model is trained on $N = 5000$ points.

