# OpenReview forum: "Neural Low-Discrepancy Sequences"
_ICLR.cc/2026/Conference — Submitted to ICLR 2026_

### Official Review · Reviewer_ksof · 2025-10-19

**Soundness:** 3
**Presentation:** 3
**Contribution:** 3
**Rating:** 6
**Confidence:** 4

**Summary:**

The paper “Neural Low-Discrepancy Sequences (NEUROLDS)” introduces the first machine learning framework for generating low-discrepancy sequences which are ordered sets of points that uniformly cover a space such that every prefix maintains low discrepancy, a property vital in numerical integration, robotics, and scientific computing. Unlike the Message-Passing Monte Carlo (MPMC) approach that only produces fixed-size low-discrepancy sets, NEUROLDS learns an extensible sequence by mapping integer indices to points using sinusoidal positional encodings and a multilayer perceptron. Training proceeds in two stages: pretraining to mimic classical sequences (e.g., Sobol’) and fine-tuning via differentiable L2-based discrepancy losses over all prefixes. Experiments show that NEUROLDS achieves substantially lower discrepancies than traditional Sobol’, Halton, and scrambled Sobol’ sequences, leading to superior performance in quasi-Monte Carlo integration (Borehole benchmark), robot motion planning (RRT), and scientific machine learning (Black–Scholes PDE). Ablation studies highlight the importance of Sobol’-based pretraining, sufficient network depth, and frequency encoding. Overall, NEUROLDS bridges number-theoretic QMC methods and neural architectures, offering a flexible, extensible, and empirically superior approach to uniform sampling.

**Strengths:**

1. NEUROLDS is the first neural framework capable of generating low-discrepancy sequences, solving a major limitation of previous learning-based methods like MPMC.

2. The combination of supervised pretraining on classical sequences (e.g., Sobol’) followed by unsupervised discrepancy minimization ensures both stability and performance.

3. NEUROLDS consistently achieves lower discrepancy and better integration accuracy across diverse applications, numerical integration, robot motion planning, and scientific ML, demonstrating versatility.

4. The index-based formulation allows sequences of arbitrary length, unlike fixed-size MPMC sets, making it practical for adaptive sampling.

5. It maintains links with classical quasi-Monte Carlo theory through kernel-based discrepancy losses and sinusoidal encodings analogous to number-theoretic digit expansions.

**Weaknesses:**

1. The method heavily relies on Sobol’ or Halton prealignment; direct training from scratch fails, indicating limited robustness in initialization.

2. Fine-tuning discrepancy losses over all prefixes scales quadratically $(\mathcal{O}(N^2))$ with sequence length, which may limit scalability to very large $N$.

3. While empirically superior, there are no formal proofs of discrepancy bounds or convergence rates as exist for classical sequences.

4. Performance depends on tuning several parameters (network depth, sinusoidal frequencies, weighting schemes), requiring careful optimization.

5.  Current formulation focuses on uniform sampling on $[0,1]^d$; extensions to non-uniform or adaptive distributions should be further discussed e.g., on the sphere [1].

[1] Quasi-Monte Carlo for 3D Sliced Wasserstein, Nguyen et al

**Questions:**

1. Can we derive the rate of convergence?

2. Can we make a randomized version of NEUROLDS?

---

> ### Author Response · Authors · 2025-11-21
> **Reply to Reviewer ksof**
>
> We thank the reviewer for appreciating the merits of our paper and their suggestions for improvement. Below, we address the concerns raised by the reviewer and thank the reviewer in advance for reading our reply.
>
> * **"The method heavily relies on Sobol’ or Halton prealignment; direct training from scratch fails, indicating limited robustness in initialization.":** We agree with the reviewer on the observation. However, we do not see it as a weakness. The underlying optimization problem is highly non-convex. Already minimizing the discrepancy of a point set (as it was done in Rusch et al, 2024 for MPMC) is a very challenging problem. Now, extending this to the setup of a sequence, where every prefix also minimizes the discrepancy makes it even more challenging. Thus, we see our pretraining approach simply as a meaningful initialization that can subsequently trained in a unsupervised manner.
>
> * **"Fine-tuning discrepancy losses over all prefixes scales quadratically with sequence length, which may limit scalability to very large $N$":** While this is true for a naive implementation, we would like to emphasize that the discrepancy of each prefix can be computed independently. Therefore, all prefix discrepancies can be evaluated in parallel, for example by dispatching them to separate GPU threads. This reduces the effective runtime back to linear in $N$.
>
> * **"While empirically superior, there are no formal proofs of discrepancy bounds or convergence rates as exist for classical sequences.":** We would like to emphasize that our approach enjoys all the theoretical
> benefits of classical low-discrepancy sequences. In particular, the discrepancy of a learned NeuroLDS sequence can be computed efficiently and then directly plugged into the extensive body of error and convergence analyses developed over the past several decades for low-discrepancy sampling methods, e.g., Koksma-Hlawka error bound inequalities. This is, in our view, one of the key advantages of our approach: we leverage AI to substantially improve upon existing constructions without discarding the well-established theoretical insights and rigorous guarantees that make low-discrepancy methods so powerful in the first place.
>
> * **"Performance depends on tuning several parameters (network depth, sinusoidal frequencies, weighting schemes), requiring careful optimization.":** We agree with the reviewer that training NeuroLDS requires tuning certain hyperparameters. However, this is inherent to virtually all machine learning methods that rely on neural networks, including standard choices such as learning rate, optimizer, network depth, and related parameters.
>
> * **"Current formulation focuses on uniform sampling on; extensions to non-uniform or adaptive distributions should be further discussed":** We thank the reviewer for this suggestion. We note that the vast majority of theoretical work on low-discrepancy sampling is formulated on the hypercube, as many practical problems can be transformed into this domain. That said, NeuroLDS can naturally be extended to non-uniform target distributions by minimizing a kernelized Stein discrepancy rather than the discrepancy measures used in this paper. We view this as a promising direction and plan to explore it in future work.
>
> We sincerely hope that we have addressed the concerns of the reviewer satisfactorily in the revised version and would kindly ask the reviewer to update their score accordingly.

---

> > ### Comment · Reviewer_ksof · 2025-11-22
> >
> > Thank you very much for the reply,
> >
> > On the convergence rate  and asymptotic uniformity, I'm still not clear why NeuroLDS  can guarantee asymptotic uniformity and convergence rate of the points set. Concretely, when the number of points go to infinity, can we prove that the star discrepancy go to 0? If yes, what is the rate in terms of the number of points?

---

> > > ### Author Response · Authors · 2025-11-25
> > > **Reply to Reviewer ksof**
> > >
> > > We thank the reviewer for appreciating our rebuttal and for their continued engagement in the scientific discussion. Below we answer the question raised in their response.
> > >
> > > We would like to clarify that NeuroLDS produces finite sequences which are low-discrepancy. A low-discrepancy sequence is defined as a (finite or infinite) sequence whose prefix discrepancies converge at the rate $\log(N)^{d-1}/N$. Under this definition, it is clear that NeuroLDS qualifies as a low-discrepancy sequence.
> > >
> > > We sincerely hope that we have addressed the remaining concern of the reviewer satisfactorily and would kindly ask the reviewer to update their score accordingly.

---

> > > > ### Comment · Reviewer_ksof · 2025-11-27
> > > >
> > > > Thank you for your response,
> > > >
> > > > I'm not sure why "it is clear that NeuroLDS qualifies as a low-discrepancy sequence". Is it based on just empirical evaluation?

---

> > > > > ### Author Response · Authors · 2025-11-27
> > > > > **Reply to Reviewer ksof**
> > > > >
> > > > > Indeed, the reviewer is correct that the low-discrepancy property can be verified empirically. We emphasize that this evaluation is computationally inexpensive and, in fact, cheaper than a single forward pass. We hope this clarification fully addresses the reviewer’s remaining concerns.

---

### Official Review · Reviewer_wMj7 · 2025-10-21

**Soundness:** 2
**Presentation:** 2
**Contribution:** 2
**Rating:** 2
**Confidence:** 4

**Summary:**

This paper presents a novel approach, NeuroLDS, for generating low-discrepancy sequences using neural networks, addressing a key limitation of prior learning-based methods like MPMC which could only produce fixed-size point sets. However, the study is marred by significant weaknesses in its experimental validation and methodological justification. The claims of superior performance are not fully supported by comprehensive comparisons, and the choice of model architecture lacks sufficient ablation analysis. Furthermore, critical practical aspects such as the computational cost and training efficiency of the proposed method are not discussed, raising concerns about its practicality.

**Strengths:**

1.The paper introduces NeuroLDS, the first machine learning-based framework for generating low-discrepancy sequences (LDS).
2. The proposed two-stage training process—supervised pre-training on classical sequences followed by unsupervised fine-tuning on a discrepancy loss—is demonstrated to be effective.

**Weaknesses:**

1.  The core discrepancy analysis is primarily conducted in 4 dimensions. The performance and scalability of NeuroLDS in both lower (1D, 2D, 3D) and higher (5D+) dimensions remain largely unexamined.
2. The set of baseline methods is not consistent across all application experiments (e.g., NM-Greedy is absent from the motion planning study). This undermines the fairness and comprehensiveness of the cross-experimental comparison.
3. The choice of a simple MLP over more established sequence-based models (e.g., Transformers, LSTMs) is not adequately justified. While an autoregressive GNN was tested and dismissed, the rationale for not exploring other powerful sequence-modeling architectures is lacking and should be supported by further discussion or ablation studies.

**Questions:**

1. The authors use a simple MLP for a sequence-generation task. There is no explanation for why more established sequence-based models, such as LSTMs or Transformers, were not considered or tested.
2. Inconsistent Baselines: The baseline methods used are not consistent across the different application experiments. For a fair and comprehensive comparison, all applied tests (e.g., numerical integration, robot planning) should evaluate the same set of baseline methods, including Sobol', Halton, NM-Greedy, and Uniform sampling.
3. The paper does not discuss or compare against several important recent works in the field, such as Atanassov's methods
4. The analysis of discrepancy is primarily conducted only in 4 dimensions. The performance and scalability of the proposed method remain unclear for other dimensions, both lower (1, 2, 3) and higher (5+).
5.  A direct and crucial comparison with Message-Passing Monte Carlo (MPMC) is lacking. Specifically, for the same number of points (N), how does the discrepancy of a NeuroLDS-generated set compare to a set generated by MPMC?

---

> ### Author Response · Authors · 2025-11-21
> **Reply to Reviewer wMj7**
>
> We thank the reviewer for appreciating the merits of our paper and their suggestions for improvement. Below, we address the concerns raised by the reviewer and thank the reviewer in advance for reading our reply.
>
> * **"The core discrepancy analysis is primarily conducted in 4 dimensions. The performance and scalability of NeuroLDS in both lower (1D, 2D, 3D) and higher (5D+) dimensions remain largely unexamined.":** We thank the reviewer for bringing this to our attention. We have now added additional plots to the revised version of our paper (Figures 10 and Figure 11). In these plots, we present the discrepancy of NeuroLDS on 1, 2, 3, and 8-dimensional cases. We note that NeuroLDS increasingly outperforms previous methods, in particular for higher dimensional settings.
>
> * **"The set of baseline methods is not consistent across all application experiments (e.g., NM-Greedy is absent from the motion planning study).":** We thank the reviewer for pointing this out. We have now added NM-Greedy to the experiment in Table 3 and to Figure 10 and Figure 11. We note that our proposed NeuroLDS still outperforms any other method. Moreover, we would like to point out that the NM-Greedy construction is too expensive for any sequence longer than 1k. For this reason, we cannot include NM-Greedy in the comparison of the other experiments.
>
>
> * **"The paper does not discuss or compare against several important recent works in the field, such as Atanassov's methods":** The authors assume the reviewer is referring to recent methods by Atanassov to improve the direction numbers for the construction of the Sobol' sequence, e.g., Atanassov \& Ivanovska (2022). However, the direction numbers introduced there do not align with the widely‐adopted QMC standard (see Joe \& Kuo, 2008), and furthermore, the Atanassov data are not publicly accessible. Our comparisons against classical Sobol' and Halton sequences are appropriate given their long-standing popularity and widespread use with demonstrated performance across decades of QMC research and applications. In principle, one could also compare against constructions such as generalized Halton sequences. However, this would require selecting high-quality digit permutations. This entails a nontrivial design choice that could confound the results and, in our opinion, distracts from the main contribution.
>
>
> * **"The authors use a simple MLP for a sequence-generation task. There is no explanation for why more established sequence-based models, such as LSTMs or Transformers, were not considered or tested.":** We have added an additional ablation in Section 3.3 where we swap out the MLP with an LSTM. While we get slightly better results using an LSTM, training the underlying model takes significantly longer: already for sequences of length 500 the MLP takes only 10 minutes to train, while the LSTM requires more than one hour. For longer sequences, this gets amplified even more until it is not feasible to use LSTMs in this context anymore.
>
> * **"A direct and crucial comparison with Message-Passing Monte Carlo (MPMC) is lacking":** We would like to point out that MPMC, as introduced in Rusch et al. (2024), is designed to generate point sets with low discrepancy for $N$ points fixed in advance. It cannot be used to generate extensible low-discrepancy sequences, which require every prefix to exhibit low discrepancy.
> To highlight this crucial distinction, we have added Figure 1 in the revised manuscript. The figure compares the discrepancy of an MPMC model trained to generate a point set of size 1024 with the discrepancy of our NeuroLDS sequence. As expected, MPMC performs very well at $N=1024$, but it fails to outperform Sobol for almost all prefixes. In contrast, NeuroLDS produces a genuine low-discrepancy sequence in which every prefix achieves substantially lower discrepancy than existing methods. Moreover, we would like to highlight that we use a completely different AI approach that is not based on GNNs anymore. This turned out to be crucial.
>
> We sincerely hope that we have addressed the concerns of the reviewer satisfactorily in the revised version and would kindly ask the reviewer to update their score accordingly.

---

> ### Comment · Reviewer_wMj7 · 2025-11-25
>
> Thank you for your feedback.
>
> We would like to clarify whether the results in Figures 10 and 11 were generated by the same model. Additionally, can the method be applied to different dimensions or sequence lengths using only a single trained model?
>
> The LSTM model appears to outperform the MLP, particularly on longer sequences. Could you provide more insight into the reasons for this improvement?
>
> If a single trained model can generalize across various dimensions and sequence lengths, training time becomes less of a concern. However, if the model's performance degrades when applied to different sequences or dimensions, the practical applicability of the method may be limited.

---

> > ### Author Response · Authors · 2025-11-25
> > **Reply to Reviewer wMj7**
> >
> > We thank the reviewer for appreciating our rebuttal and for their continued engagement in the scientific discussion. Below we answer the questions raised in their response.
> >
> > * **"We would like to clarify whether the results in Figures 10 and 11 were generated by the same model. Additionally, can the method be applied to different dimensions or sequence lengths using only a single trained model?":** The models shown in Figure 10 and Figure 11 are not derived from the same trained model. Although NeuroLDS can in principle be generalized to different dimensions by training on a discrepancy measure that emphasizes lower-dimensional coordinate projections and then using these during inference, our focus here was on extending the original MPMC approach from point sets to sequences. For this reason, we adopted the same training and evaluation protocol as in the original MPMC work.
> >
> > * **"The LSTM model appears to outperform the MLP, particularly on longer sequences. Could you provide more insight into the reasons for this improvement?":** Since the LSTM also operates on the indices, there is no new training framework compared to our proposed MLP. In particular, the LSTM is not autoregressively unrolled, as this does not work in practice. For this reason, we believe that the slight performance difference arises from architectural differences, with the LSTM being slightly more expressive. However, we also note that training the LSTM takes significantly longer and yields only negligibly better sequences.
> >
> > We sincerely hope that we have addressed the remaining concerns of the reviewer satisfactorily and would kindly ask the reviewer to update their score accordingly.

---

### Official Review · Reviewer_92S7 · 2025-10-27

**Soundness:** 3
**Presentation:** 2
**Contribution:** 3
**Rating:** 4
**Confidence:** 4

**Summary:**

This study proposes a new method to define collocation points, in particular, to construct low-discrepancy sequences, with which we can achieve good accuracy even when truncated to a moderate number of points, using a neural network. The neural network receives an index and returns a point. If the integrand lies in an RKHS, the error between the true integral and the finite-point approximation is bounded by the product of the integrand's variation and the discrepancy that depends on the chosen points. Since the discrepancy can be computed from the kernel, we can choose points that minimize it. Therefore, the neural network is trained to output a sequence that minimizes equation (2) for every N.

**Strengths:**

As stated in the introduction, the approach is clearly supported by theory, and uses a neural network to resolve the intractable portion, extending the reach of theoretical research.

**Weaknesses:**

This paper is a direct extension of Rusch et al. (2024). Equation (2) in this paper corresponds to equation [2] in Rusch et al. (2024), and the evaluation protocol is also an extension of theirs. Although there is some improvements in the neural network design, that part does not seem essential. For example, if this were a workshop submission supplementing a main-conference paper, such a proposal might be sufficient. However, it does not appear to reach the level of originality expected for the conference main track.

The paper claims that N need not be fixed, but in practice. one must train a GNN on sequences up to N, so in that sense N cannot be freely set.

The comparisons are mostly in terms of accuracy, but accuracy naturally improves as the number of points increases. What matters in point selection is the number of points (that is, the computational cost) required to achieve the same accuracy. According to Figure 2 and related results, the efficiency gain appears at most about a factor of two. Once we account for training the neural network, the total computational cost may not be favorable.

Introducing $\gamma$ makes the method integrand-dependent, which implies retraining for each problem and, in turn, an increase in total computational cost. A fair comparison on this point is also needed.

The loss is defined as an average over all N, but this seems misaligned with the definition of low-discrepancy sequences. An approach that would align with the definition is: optimize a GNN for N=1; once that is done, fix the output for N=1 and then optimize the GNN for N=2; once that is done, proceed to N=3; and so on.

For this purpose, it may not be necessary to use a GNN at all, right? If the values of the sequence can simply be stored in a table, what is the reason to generate them with a GNN?

QMC methods allow collocation points randomized, which provides variance estimates. Does the inability to randomize become a disadvantage for the proposed method? Existing methods can be randomized; for the proposed method, one could randomize the sequences for pre-training and train GNNs multiple times, then examine performance variance. Such an evaluation seems necessary.

Empirically, the method shows good accuracy, but theoretical guarantees are limited. Existing methods have known orders. What about this method? For example, using similar functions while varying the dimension and the number of points, one could regress a two-dimensional plot to estimate the order at least approximately. My guess is that the improvement is a constant-factor one.

Minor Comments:

The citation style is odd. In places where citep should be used, citet is used, so it is unclear whether the citation is part of the sentence or not.

The theoretical core is written in the introduction, and the Methods section contains a long explanation of related work, after which the proposal begins. The overall structure could be improved.

**Questions:**

See Weaknesses.

---

> ### Author Response · Authors · 2025-11-21
> **Reply to Reviewer 92S7 Part 1 of 2**
>
> We thank the reviewer for appreciating the merits of our paper and their suggestions for improvement. Below, we address the concerns raised by the reviewer and thank the reviewer in advance for reading our reply.
>
> * **"This paper is a direct extension of Rusch et al. (2024).":** We respectfully disagree with the reviewer on this point. MPMC, as introduced in Rusch et al. (2024), is designed to generate point sets with low discrepancy. It cannot be used to generate low-discrepancy sequences, which require every prefix to exhibit low discrepancy.
> To highlight this crucial distinction, we have added Figure 1 in the revised manuscript. The figure compares the discrepancy of an MPMC model trained to generate a point set of size 1024 with the discrepancy of our NeuroLDS sequence. As expected, MPMC performs very well at $N=1024$, but it fails to outperform Sobol for almost all prefixes. In contrast, NeuroLDS produces a genuine low-discrepancy sequence in which every prefix achieves substantially lower discrepancy than existing methods. Moreover, we would like to highlight that we use a completely different AI approach that is not based on GNNs anymore. This turned out to be crucial.
>
> * **"$N$ cannot be freely set":** We thank the reviewer for raising this important point. The purpose of low-discrepancy sequences is to generate sampling points such that every prefix (i.e., every initial subsequence) exhibits low discrepancy. We agree that generalization to any $N$ is desirable. However, we do not view the lack of such universal generalization as a practical limitation of our approach.
> In most real-world applications, the required number of sampling points is known in advance. Low-discrepancy sampling is typically employed only when evaluating the underlying integrand or observable is extremely costly; otherwise, standard Monte Carlo with uniform random numbers is sufficient. Because we can train NeuroLDS for any chosen $N$, one can simply train a sequence with an $N$ that comfortably exceeds the anticipated number of samples.
> Moreover, NeuroLDS sequences can be trained entirely offline. That is, we can train a sequence with a very large $N$ once, and then reuse it across downstream tasks without any additional computational cost. This is precisely what demonstrate in our experiments.
>
> * **"What matters in point selection is the number of points (that is, the computational cost) required to achieve the same accuracy":** We fully agree with the reviewer on this and have added a new experimental evaluation to the revised version of our paper. In Section 3.2.2 of the revised version we now present the required number of points for every other method to match the performance of NeuroLDS on the robotics motion planning task. We can see that the competing methods require at least 1.55 to 2.5 times more sampling points to reach the same number of accuracy.
>
> * **"Once we account for training the neural network, the total computational cost may not be favorable":** We respectfully disagree with the reviewer on this point. First, we emphasize that NeuroLDS can be trained offline and then reused for any downstream task that requires low-discrepancy sampling. This allows us to amortize the training cost over many applications. Second, low-discrepancy sampling is typically employed only when evaluating the underlying integrand or observable is very expensive; otherwise, standard Monte Carlo methods suffice. In such settings, any reduction in the number of required samples to achieve a given accuracy is extremely valuable. Consequently, even moderate improvements in discrepancy directly translate into significant practical gains.
>
> * **"Introducing $\gamma$ makes the method integrand-dependent, which implies retraining for each problem and, in turn, an increase in total computational cost.":** We fully agree with the reviewer on that. Indeed, adjusting for specific weights of the underlying dimensions biases our sampling points toward specific applications. That being said, this is a particular feature of optimization-based low-discrepancy sampling and is not possible with most classical constructions. Thus, any method that incorporates these weights has to be specifically optimized and constructed for the specific $\gamma$. We finally note that many applied problems exhibit this same low-dimensional structure, e.g., option pricing in quantitative finance, so the cost of training NeuroLDS for a specific choice of $\gamma$ can be amortized across multiple related implementations.

---

> > ### Author Response · Authors · 2025-11-21
> > **Reply to Reviewer 92S7 Part 2 of 2**
> >
> > * **"optimize a GNN for N=1; once that is done, fix the output for N=1 and then optimize the GNN for N=2; once that is done, proceed to N=3; and so on.":** The approach described by the reviewer would be an autoregressive extension of MPMC. It is very interesting that the reviewer suggests this, because this was our initial idea of the method. Unfortunately, this did not yield good result and was too expensive to train for large $N$. The approach we finally adopted in our manuscript enabled a fast training over an arbitrary length. We have highlighted this in our revised version and thank the reviewer again for bringing it up.
> >
> > * **"it may not be necessary to use a GNN at all, right?":** We would like to clarify that at no point in our method we are using GNNs. We are simply parameterizing the index-to-points mapping via a simple MLP. This approach enables the fast training of NeuroLDS for any desired sequence length.
> >
> > * **"for the proposed method, one could randomize the sequences for pre-training and train GNNs multiple times":** We thank the reviewer for this excellent suggestion. Indeed, the suggested approach by the reviewer is a viable approach for randomizing our proposed NeuroLDS. Please note that this is exactly what we do for the robotics motion planning task in order to get average success-rate results. We have further emphasized this in the revised version, and thank the reviewer again for the suggestion.
> >
> > * **"Existing methods have known orders. What about this method?":** Classical low-discrepancy sequences are designed so that the discrepancy converges at the rate $log(N)^d/N$. However, none of the existing constructions achieve this rate with an optimal constant. The underlying reason is that identifying a truly optimal sequence requires solving a highly non-convex optimization problem. This is exactly where neural networks effective, as they provide a powerful framework for handling such complex non-convex objectives.
> > Our method can be viewed as learning a low-discrepancy sequence that not only attains the correct asymptotic scaling but also achieves a nearly optimal constant. This constant has substantial practical impact: for a fixed $N$, our learned sequences can reduce the discrepancy for instance by a factor of two compared to classical constructions, which translates directly into significantly improved performance in downstream applications.
> >
> > * We have incorporated the minor comments and fixed the citation style. We thank the reviewer for bringing this to our attention.
> >
> >
> > We sincerely hope that we have addressed the concerns of the reviewer satisfactorily in the revised version and would kindly ask the reviewer to update their score accordingly.

---

> > > ### Comment · Reviewer_92S7 · 2025-11-22
> > >
> > > Thank you very much for your detailed replies. I also apologize for my misunderstanding about GNNs and randomization. I have also understood the theoretical justification. The convergence rate is determined by the design of the objective function, and your method optimizes the constant factor in this rate.
> > >
> > > Other reviewers point out that the method is limited to low dimensions, but I do not consider this to be a problem. Existing low-discrepancy sequences are also specialized to low-dimensional spaces, so this is not a disadvantage of the proposed method but rather an issue of the field.
> > >
> > > > because this was our initial idea of the method. Unfortunately, this did not yield good result and was too expensive to train for large N.
> > >
> > > This observation is very interesting. It suggests that, when N is small, the method may behave as if it is overfitting to that particular N. In that sense, it may not be appropriate to refer to the resulting points as low-discrepancy sequences. For example, the first 5000 points of the points obtained by training with N = 10000 may differ from the points obtained by training with N = 5000. I am curious how large the performance difference between these two cases would be. From a practical aspect, there is no problem if using 5000 points out of N = 10000 still yields good performance.
> > >
> > > I am not asking for additional experiments. Rather, I am interested in the conceptual question of whether what your method produces can be regarded as low-discrepancy sequences in the same sense as existing methods.
> > >
> > > The need for hyperparameter tuning, which another reviewer pointed out, would not be an issue for general NN research, but in the context of collocation point selection, it can clearly be regarded as a disadvantage.
> > >
> > > For these reasons, my current impression is that, "although the empirical performance is undoubtedly good, the theoretical justification and the positioning of the method as a low-discrepancy sequence remain unclear".
> > >
> > > Hence, I keep my score at 4.

---

> > > > ### Author Response · Authors · 2025-11-25
> > > > **Reply to Reviewer 92S7**
> > > >
> > > > We thank the reviewer for appreciating our rebuttal and for their continued engagement in the scientific discussion. Below we address the two main points raised in their response.
> > > >
> > > > * **"the first 5000 points of the points obtained by training with N = 10000 may differ from the points obtained by training with N = 5000. I am curious how large the performance difference between these two cases would be.":** The reviewer is correct that a NeuroLDS model trained on 1000 points and one trained on 5000 points produce slightly different low-discrepancy sequences, even when considering only the first 1000 points of the 5000-point model. This is expected, since the underlying optimization problems differ: the 5000-point model must ensure low discrepancy for a larger set of prefixes. However, the resulting performance difference in terms of discrepancy is negligibly small. To support this claim empirically, we have added a new plot (Figure 12) in the revised version of the paper. It compares the two NeuroLDS sequences trained on 1k and 5k points, respectively, and shows that both achieve low discrepancies, with only minimal differences between them.
> > > >
> > > > * **"the theoretical justification and the positioning of the method as a low-discrepancy sequence remain unclear":** We respectfully disagree with the reviewer on this point. A low-discrepancy sequence is defined as a (finite or infinite) sequence whose prefix discrepancies converge at the rate $\log(N)^{d-1}/N$. Under this definition, it is clear that NeuroLDS qualifies as a low-discrepancy sequence.
> > > >
> > > > We hope that this addresses the remaining concerns and that the reviewer will consider raising their score.

---

### Official Review · Reviewer_FumR · 2025-10-31

**Soundness:** 2
**Presentation:** 2
**Contribution:** 2
**Rating:** 2
**Confidence:** 4

**Summary:**

The authors employ a learning-based approach to generate quasi-random samples. However, the main concern lies in the fact that this approach relies on classical quasi-random methods (such as Sobol’ or Halton sequences) to generate the training data. This creates a dilemma: the performance of the trained model inherently depends on the quality of the training samples. If the proposed method truly works as claimed, it would imply that one could achieve the same integration accuracy using only 100 samples as with 10,000 samples without any additional cost, which seems questionable.

**Strengths:**

No.

**Weaknesses:**

1. Lack of Distinct Advantage: Unlike other learning-based methods used in scientific applications, the proposed framework does not demonstrate a clear advantage compared to classical methods. For instance, Physics-Informed Neural Networks (PINNs) are valuable for solving high-dimensional or inverse problems compared to classical numerical methods. In contrast, it is unclear what specific capability the proposed method offers that cannot already be achieved by classical approaches.
2. Absence of Theoretical Foundations: The work provides no theoretical guarantees or analysis (e.g., discrepancy bounds or convergence rates). This is a critical shortcoming for a method that aims to replace well-established number-theoretic constructions, making it impossible to evaluate its reliability or general behavior beyond limited empirical observations.
3. Lack of High-Dimensional Validation: The experiments are primarily conducted in low-dimensional settings (e.g., 2d or 4d). The absence of results in high-dimensional cases (e.g., $d > 10$) significantly undermines the claimed generality and applicability of the method, particularly for modern problems in machine learning and computational finance.

**Questions:**

see the Weaknesses part

---

> ### Author Response · Authors · 2025-11-21
> **Reply to Reviewer FumR**
>
> We thank the reviewer for their suggestions for improvement. Below, we address the concerns raised by the reviewer and thank the reviewer in advance for reading our reply.
>
> * **"Lack of Distinct Advantage":** We thank the reviewer for raising this point. Our approach can be viewed as a hybrid ansatz. On the one hand, we ensure that our method retains the theoretical benefits of classical low-discrepancy sampling, including the corresponding convergence guarantees. On the other hand, we leverage AI to substantially improve upon existing constructions.
> Classical low-discrepancy sequences are designed so that the discrepancy converges at the rate $log(N)^d/N$. However, none of the existing constructions achieve this rate with an optimal constant. The underlying reason is that identifying a truly optimal sequence requires solving a highly non-convex optimization problem. This is exactly where neural networks become effective, as they provide a powerful framework for handling such complex non-convex objectives.
> Our method can be viewed as learning a low-discrepancy sequence that not only attains the correct asymptotic scaling but also achieves a nearly optimal constant. This constant has substantial practical impact: for a fixed $N$, our learned sequences can reduce the discrepancy for instance by a factor of two compared to classical constructions, which translates directly into significantly improved performance in downstream applications. We also appreciate the reviewer’s analogy to PINNs. Extending this analogy, our approach can be viewed as using AI to enhance the efficiency and accuracy of classical numerical methods without discarding their theoretical guarantees. In contrast, PINNs are fully AI-based and therefore do not provide rigorous convergence or error analyses.
>
> * **"Absence of Theoretical Foundations":** We would like to emphasize that our approach enjoys all the theoretical benefits of classical low-discrepancy sequences. In particular, Koksma-Hlawka inequalities bound the error of numerically estimating an expectation by the product of the discrepancy of the sampling nodes and a notion of variation of the integrand. Thus, a smaller discrepancy automatically implies a smaller error.
> This is, in our view, one of the key advantages of our approach: we leverage AI to substantially improve upon existing constructions without discarding the well-established theoretical insights and rigorous guarantees that make low-discrepancy methods so powerful in the first place.
>
> * **"Lack of High-Dimensional Validation":** That is an excellent point raised by the reviewer. While low-discrepancy sequences have been applied successfully in very high-dimensional settings, sometimes with hundreds or even thousands of dimensions, these applications share an important characteristic: the effective dimensionality of the underlying problem is typically very small, often only two to four dimensions. In other words, although the ambient dimension is large, the integrand or problem structure reduces the intrinsic dimensionality to a low range.
> For this reason, the performance of our proposed NeuroLDS is fully and appropriately assessed by the low- to medium-dimensional problems examined in this manuscript. This claim is further supported by the theoretical lower bound on discrepancy, which behaves as $log(N)^d/N$. Asymptotically, the dominant term is $1/N$, while the dependence on $d$ appears in the pre-asymptotic constant. Consequently, when $d$ becomes very large, the constant deteriorates to the point where low-discrepancy methods no longer outperform classical Monte Carlo. Thus, for extremely high-dimensional problems, LDS methods—including ours—are not expected to offer practical advantages.
>
> We sincerely hope that we have addressed the concerns of the reviewer satisfactorily in the revised version and would kindly ask the reviewer to update their score accordingly.

---

> ### Comment · Reviewer_FumR · 2025-11-27
>
> The biggest concern: the training data is from classical quasi-random methods (see section 2.2). This creates a dilemma: the performance of the trained model inherently depends on the quality of the quasi-random samples. It is hard to say this can improve the integration accuracy. The experimental results in Table 1 (also other results) indicate that this method does not have   obvious advantage.

---

> > ### Author Response · Authors · 2025-12-01
> > **Reply to Reviewer FumR**
> >
> > We thank the reviewer for their follow-up remark. While we appreciate the continued feedback, we have to disagree with the reviewer on their two comments:
> > * **"the performance of the trained model inherently depends on the quality of the quasi-random samples":** Existing low-discrepancy sequences such as Sobol and Halton are only used to initialize NeuroLDS during the pre-training stage. This can be interpreted as similar to standard initialization schemes such as Kaiming or Xavier initialization for MLPs. The trained NeuroLDS sequences diverge substantially from these initializations, as clearly shown in Figure 1, Figure 3, Figure 6, Figure 10, and Figure 11. All of these figures unambiguously demonstrate that NeuroLDS produces low-discrepancy sequences that differ markedly from Sobol points. This is further evident in Figure 6, where we plot Sobol and NeuroLDS points in two dimensions as a scatter plot. The two sets of points are visually and structurally distinct.
> > * **"The experimental results in Table 1 (also other results) indicate that this method does not have obvious advantage.":** We strongly disagree with the reviewer on this point. Table 1 represents one of the most challenging benchmarks for QMC methods. The fact that NeuroLDS outperforms all other low-discrepancy sequences for most choices of $N$ clearly demonstrates its advantage. In addition, we present demanding experiments in robotics motion planning and scientific machine learning and show that NeuroLDS consistently outperforms all previous low-discrepancy methods on every benchmark we consider. This provides strong evidence for the advantage of NeuroLDS.

---

### Official Review · Reviewer_w9Dh · 2025-11-02

**Soundness:** 3
**Presentation:** 3
**Contribution:** 2
**Rating:** 4
**Confidence:** 3

**Summary:**

This paper considers the problem of approximating a high-dimensional integral using Monte-Carlo methods. For simplicity, the authors assume that the function is supported on the hypercube $[0, 1]^d$, and the goal is to generalize data points that are as distributed as uniformly as possible. Since the approximation error is upper bounded by the kernel discrepancy of the point sequence $\lbrace X_i \rbrace_{i=1}^{N}$, a natural goal would be to generate a sequence whose kernel discrepancy is as low as possible. To generate such a low-discrepancy sequence (LDS, to be differentiated from a low-discrepancy set), the authors propose NeuroLDS, an L-layer neural network that maps an integer index $i$ to a point $X_i \in [0,1]^d$. This "index-driven" architecture is inspired by classical LDS like Sobol' and Halton sequences. The authors also propose a two-stage optimization procedure for training this neural network and conduct extensive numerical experiments to demonstrate the superior performance of NeuroLDS as compared to classical methods.

**Strengths:**

Overall I think this is an interesting paper that makes contribution to an active research area. The main results are also clearly written.

**Weaknesses:**

(i) I think the goal of NeuroLDS is to generate a low-discrepancy sequence, not just a low-discrepancy set of points. One would expect to generate an infinite sequence whose all N-prefixes are of low-discrepancy, is it correct? If so, I am curious how the trained network generalizes to indices $i > N$.
(ii) The experiments are conducted in relatively low dimensions (i.e., $d=2, 4, 8$). However, the key challenge for classical LDS is the "curse of dimensionality." It is still unclear if NeuroLDS offers any fundamental advantage in truly high-dimensional settings, for example, $d > 20$?
(iii) I understand that this is a purely experimental paper, but it would be better to have some theory, even non-rigorous heuristic calculations. What is the intuition behind the empirical observation that NeuroLDS outperforms classical LDS?

**Questions:**

Please see "Weaknesses" section. This paper is well written so I do not have minor comments on the writting.

---

> ### Author Response · Authors · 2025-11-21
> **Reply to Reviewer w9Dh**
>
> We thank the reviewer for appreciating the merits of our paper and their suggestions for improvement. Below, we address the concerns raised by the reviewer and thank the reviewer in advance for reading our reply.
>
> * **"Generalization for $i > N$":** We thank the reviewer for this question. The purpose of low-discrepancy sequences is to generate sampling points such that every prefix (i.e., every initial subsequence) exhibits low discrepancy. We agree that generalization to any $N$ is desirable. However, we do not view the lack of such universal generalization as a practical limitation of our approach.
> In most real-world applications, the required number of sampling points is known in advance. Low-discrepancy sampling is typically employed only when evaluating the underlying integrand or observable is extremely costly; otherwise, standard Monte Carlo with uniform random numbers is sufficient. Because we can train NeuroLDS for any chosen $N$, one can simply train a sequence with an $N$ that comfortably exceeds the anticipated number of samples.
> Moreover, NeuroLDS sequences can be trained entirely offline. That is, we can train a sequence with a very large $N$ once, and then reuse it across downstream tasks without any additional computational cost. This is precisely what we demonstrate in our experiments.
>
>
> * **"It is still unclear if NeuroLDS offers any fundamental advantage in truly high-dimensional settings, for example, $d>20$":** That is an excellent question. While low-discrepancy sequences have been applied successfully in very high-dimensional settings, sometimes with hundreds or even thousands of dimensions, these applications share an important characteristic: the effective dimensionality of the underlying problem is typically very small, often only two to four dimensions. In other words, although the ambient dimension is large, the integrand or problem structure reduces the intrinsic dimensionality to a low range.
> For this reason, the performance of our proposed NeuroLDS is fully and appropriately assessed by the low- to medium-dimensional problems examined in this manuscript. This claim is further supported by the theoretical lower bound on discrepancy, which behaves as $log(N)^d/N$. Asymptotically, the dominant term is $1/N$, while the dependence on $d$ appears in the pre-asymptotic constant. Consequently, when $d$ becomes very large, the constant deteriorates to the point where low-discrepancy methods no longer outperform classical Monte Carlo. Thus, for extremely high-dimensional problems, LDS methods—including ours—are not expected to offer practical advantages.
>
> * **"What is the intuition behind the empirical observation that NeuroLDS outperforms classical LDS?":** Classical low-discrepancy sequences are designed so that the discrepancy converges at the rate $log(N)^d/N$. However, none of the existing constructions achieve this rate with an optimal constant. The underlying reason is that identifying a truly optimal sequence requires solving a highly non-convex optimization problem. This is exactly where neural networks effective, as they provide a powerful framework for handling such complex non-convex objectives.
> Our method can be viewed as learning a low-discrepancy sequence that not only attains the correct asymptotic scaling but also achieves a nearly optimal constant. This constant has substantial practical impact: for a fixed $N$, our learned sequences can reduce the discrepancy for instance by a factor of two compared to classical constructions, which translates directly into significantly improved performance in downstream applications.
>
> We sincerely hope that we have addressed the concerns of the reviewer satisfactorily in the revised version and would kindly ask the reviewer to update their score accordingly.

---

### Author Response · Authors · 2025-11-21
**Reply to all the reviewers**

At the outset, we would like to thank all five reviewers for their thorough and patient reading of our article. Their fair criticism and constructive suggestions have enabled us to improve the quality of our article. A revised version of the article is uploaded. We proceed to answer the points raised by each of the reviewers individually, below. We would also like to point out that all the references to page numbers, sections, figures, tables, equation numbers and references, refer to those in the revised version. We hope that based on the answers to their questions and the improvements made to the revised version, the reviewers will increase their scores.

---

### Meta-Review · Area_Chair_7Re2 · 2026-01-07

**Summary:**

In their paper, the authors propose Neural Low-Discrepancy Sequences (NeuroLDS), a learning-based framework that uses neural networks to generate point sequences with low discrepancy for numerical integration and Monte Carlo–type applications. Reviewers generally agreed that the idea of learning quasi–Monte Carlo–style sequences is interesting and that the paper explores a timely intersection of numerical integration and deep learning. The empirical results suggest that the proposed approach can outperform standard random sampling and, in some cases, match classical low-discrepancy sequences on selected benchmarks. However, several important weaknesses were raised that were not sufficiently addressed in the rebuttal.

*Limited evaluation scope and modest gains:*
Reviewer concerns focused on the limited range of integration tasks and dimensional settings considered in the experiments. Most evaluations are conducted on relatively low-dimensional or synthetic benchmarks, where classical low-discrepancy sequences are already well understood. Moreover, while NeuroLDS sometimes improves over random sampling, its gains over strong hand-designed baselines (e.g., Sobol or Halton sequences) are often modest and inconsistent, making it unclear whether the added complexity of training neural generators is justified.

*Fairness and clarity of the experimental comparison:*
Some reviewers remained unconvinced about the fairness of the experimental setup. In particular, questions were raised about hyperparameter tuning for NeuroLDS versus classical baselines, the sensitivity of results to training choices, and whether the baselines were implemented and tuned optimally.

*Lack of theoretical foundation:*
A central motivation of the paper is to produce sequences with provably good discrepancy properties. However, the paper does not provide theoretical guarantees on discrepancy, convergence rates, or generalization beyond the training distribution. Reviewers noted that even partial theoretical analysis—such as bounds under simplifying assumptions or connections to known discrepancy measures—would significantly strengthen the contribution.

*Unclear practical relevance.*
While the idea is conceptually appealing, reviewers questioned the practical impact of NeuroLDS. Classical low-discrepancy sequences are deterministic, cheap to generate, and well studied, whereas NeuroLDS introduces nontrivial training cost and additional complexity. The paper does not convincingly demonstrate scenarios where learned sequences provide a clear advantage that outweighs these costs, particularly in high-dimensional or real-world integration problems.

*Recommendation:*
Given the limited empirical scope, modest improvements over strong baselines, unresolved concerns about evaluation fairness, and the lack of theoretical guarantees supporting the core claims, I recommend rejecting the paper in its current form. I encourage the authors to address the reviewers’ suggestions by strengthening the theoretical foundations, broadening and deepening the experimental evaluation, and clarifying the practical regimes where learned low-discrepancy sequences offer clear benefits.

**Reviewer Concerns:**

Please refer to the summary.

**Reviewer Scores:**

Please refer to the summary.

---

### Decision · Program_Chairs · 2026-01-26

Reject